# Higher-group symmetry of (3+1)D fermionic $\mathbb{Z}_2$ gauge theory: logical CCZ, CS, and T gates from higher symmetry

Maissam Barkeshli[1], Po-Shen Hsin[2], Ryohei Kobayashi[1]

April 9, 2024

[1]Department of Physics, Condensed Matter Theory Center, and Joint Quantum Institute, University of Maryland, College Park, Maryland 20742, USA
[2]Mani L. Bhaumik Institute for Theoretical Physics, Department of Physics and Astronomy, University of California, Los Angeles, CA 90095, USA

**Abstract**

It has recently been understood that the complete global symmetry of finite group topological gauge theories contains the structure of a higher-group. Here we study the higher-group structure in (3+1)D $\mathbb{Z}_2$ gauge theory with an emergent fermion, and point out that pumping chiral $p + ip$ topological states gives rise to a $\mathbb{Z}_8$ 0-form symmetry with mixed gravitational anomaly. This ordinary symmetry mixes with the other higher symmetries to form a 3-group structure, which we examine in detail. We then show that in the context of stabilizer quantum codes, one can obtain logical CCZ and CS gates by placing the code on a discretization of $T^3$ (3-torus) and $T^2 \rtimes_{C_2} S^1$ (2-torus bundle over the circle) respectively, and pumping $p + ip$ states. Our considerations also imply the possibility of a logical $T$ gate by placing the code on $\mathbb{RP}^3$ and pumping a $p + ip$ topological state.

# 1  Introduction

A key distinguishing feature of topologically ordered phases of matter is the universal structure that arises from their topological defects. For example, anyons in (2+1)D topologically ordered phases of matter [1] can be thought of as codimension-2 topological defects, which exhibit universal braiding and fusion properties that form the structure of a unitary modular tensor category (UMTC) [2, 3, 4, 5]. Over the last 10-15 years, it has been understood that, in addition to anyons, (2+1)D topological phases of matter host other classes of codimension-1 and 2 topological defects with non-trivial universal properties [6, 7, 8, 9, 10, 11, 12, 13, 14, 15, 16, 17, 18]. These include topologically distinct classes of gapped codimension-1 defects, and codimension-2 domain walls between distinct codimension-1 defects. Altogether, the universal properties of these topological defects of varying codimension is expected to form the structure of a unitary fusion 2-category [19, 20]. In higher dimensions, $(d+1)$D topologically ordered phases of matter can host topological defects of varying codimension, which are expected to form a unitary fusion $d$-category.

In modern language, the topological defects are interpreted as defining a "higher symmetry" of the corresponding topological quantum field theory (TQFT) [21, 22, 23, 24]. From the perspective of quantum many-body systems, the topological defects can be understood as defining *emergent* higher symmetries of the ground state subspace of a topological phase of matter. The higher symmetries arise from sweeping the topological defects through lower dimensional submanifolds of the space on which the system is defined. They are in general *emergent* symmetries because they keep the ground state subspace invariant, but do not necessarily commute with the microscopic many-body Hamiltonian.

A special class of topological defects are invertible, which means they possess an "inverse" defect with which they fuse to the trivial defect. The invertible codimension-$k$ defects are associated with $(k-1)$-form higher symmetries [21]. The complete algebraic structure defined by the invertible defects defines an invertible subcategory of the higher fusion category of defects [25]. This invertible subcategory forms the mathematical structure of a higher-group. Such higher-group symmetry has appeared in the study of (2+1)D topological phases [17, 26, 27] and in quantum field theories more generally [28, 29, 30, 31, 32, 33, 32, 34, 35, 36, 37, 38, 39].

Recently, it has been observed that finite group gauge theories in $(d+1)$D generally possess a $d$-group symmetry [33, 40], which arises by considering gauged invertible topological phases [41, 42, 43, 44, 45, 46, 47, 48] decorated on lower dimensional submanifolds, in addition to the usual electric and magnetic defects. The interaction between the magnetic defects and the gauged invertible phases causes the higher form symmetry to mix into a non-trivial higher group.

These topological defects and corresponding emergent higher symmetries have two important applications. One is in characterizing and classifying symmetry-enriched topological phases of matter; roughly speaking, topological

phases of matter with a symmetry group $G$ can be characterized by a certain map from $G$ to the higher fusion category that defines the emergent higher symmetry [17, 49].

The second major application, which is the primary focus of this paper, is for fault-tolerant quantum computation. Topologically ordered phases of matter can in principle provide a physical substrate for fault-tolerant quantum computation [50, 51, 52, 1]. Moreover, most, if not all quantum error correcting codes can be understood in terms of topologically ordered states of matter with qubits defined on an appropriate cellulation of some manifold [50, 53, 54, 55, 56, 57]. The code subspace corresponds to the topologically degenerate ground state subspace of the parent Hamiltonian. It is well-known that one way to obtain fault-tolerant logical gates is through mapping class group operations, e.g. braids and Dehn twists in (2+1)D, which can be viewed as sweeping geometric defects through the system. The emergent higher symmetry operations corresponding to sweeping topological defects of varying codimension give rise to another class of fault-tolerant logical gates on the code subspace [58, 59, 60, 61, 33]. This insight has led to new ways of implementing logical gates even in the well-studied $\mathbb{Z}_2$ toric codes [40, 62]. Remarkably, Ref. [33] showed that the non-trivial algebraic relationships among various logical gates in (3+1)D $\mathbb{Z}_2$ toric codes can be understood as fundamentally arising from the higher group structure defined by the topological defects: logical gates that correspond to sweeping codimension-$k$ defects have group commutators that produce logical gates corresponding to higher codimension defects. As such, understanding deeply the nature of topological defects and corresponding emergent higher symmetries is crucial for understanding the possible fault-tolerant logical gates that a given quantum error correcting code admits.

In this paper, we focus on the case of $\mathbb{Z}_2$ gauge theory in (3+1)D where the $\mathbb{Z}_2$ electric charge is a fermion, which we refer to as fermionic $\mathbb{Z}_2$ gauge theory. Such a theory can arise from the fermionic toric code model [63, 64, 65]. The theory can also arise as a description of superconductors, where the condensation of Cooper pairs breaks the dynamical electromagnetic $U(1)$ gauge group to $\mathbb{Z}_2$ subgroup [66]. It can also arise in color superconducting phase of non-Abelian gauge theory [67].

The (3+1)D fermionic $\mathbb{Z}_2$ gauge theory hosts a codimension-1 topological defect, which arises by decorating codimension-1 submanifolds with a $p + ip$ invertible topological phase prior to gauging $\mathbb{Z}_2$ fermion parity.[1] As we discuss, this topological defect defines a $\mathbb{Z}_8$ emergent 0-form symmetry. In addition to this $\mathbb{Z}_8$ 0-form symmetry, the theory also has a 2-form symmetry generated by the Wilson line, and 1-form symmetries generated by the magnetic flux and gauged Kitaev chains [68, 33]. We discover that these symmetries form a 3-group, which corresponds to the fact that the generators of the symmetries obey intricate commutation relations. By exploiting submanifolds with different topologies that support the symmetry generators, we discover various logical gates in the fermionic $\mathbb{Z}_2$ toric code. This includes a fault-tolerant logical CCZ gate when the system is defined on a discretization of the 3-torus $T^3$ [69], and a new fault-tolerant Controlled-S gate when the system is placed on $T^2 \rtimes S^1$. Our considerations also imply the possibility of a T (i.e. $\pi/8$) gate when the system is placed on a discretization of $\mathbb{RP}^3$.

This work is organized as follows. In section 2, we investigate the symmetries in $\mathbb{Z}_2$ gauge theory with an emergent fermion in (3+1)D. Using such symmetries, we derive the corresponding logical gates in section 3. From section 4 to section 6, we discuss various examples of logical gates obtained by taking different topologies of the symmetry generators. In Appendix A, we derive the logical gate for the 1-form symmetry generated by surface operator decorated with Kitaev chain. In Appendix B, we derive the logical gate for the symmetry generated by the domain wall decorated with p+ip topological superconductor. In Appendix C we summarize the details for pumping a Chern insulator. In Appendix D we review the fermionic toric code lattice model. In Appendix E we give details for the partial rotation used in pumping the p+ip phase.

# 2    3-group symmetry of (3+1)D $\mathbb{Z}_2$ gauge theory with emergent fermion

In this section, we describe the higher symmetries of the (3+1)D fermionic $\mathbb{Z}_2$ gauge theory, their 3-group structure, and the associated 't Hooft anomaly. In the fermionic $\mathbb{Z}_2$ gauge theory, the Wilson line is a fermion instead of a boson as in ordinary $\mathbb{Z}_2$ gauge theory. We can view the gauge field as a "dynamical spin structure," where magnetic flux corresponds to defects in the spin structure.[2] The theory can be obtained from the trivial fermionic invertible phase by gauging the fermion parity symmetry without additional local counterterm. After this gauging process, the fermion becomes topologically non-trivial, meaning there are no local operators that create an isolated fermion.

We can also describe the theory in terms of the low energy properties of the fermionic toric code model in (3+1)D on cubic lattice [63] (see also [70] for similar construction for $U(1)$ gauge theory with electrons). The vertex term is

---

[1]In this paper, a p+ip superconductor refers to a fermionic invertible phase with $\mathbb{Z}_2^f$ symmetry carrying chiral central charge $c_- = 1/2$, where no spontaneous $U(1)$ symmetry breaking is implied. Also, in this paper a Chern insulator refers to a fermionic invertible phase with $\mathbb{Z}_2^f$ symmetry carrying $c_- = 1$, where $U(1)$ global symmetry is not implied.

[2]More precisely, magnetic flux defects can be understood as a cutting out a neighborhood of a codimension-2 submanifold and changing the spin structure (fermion boundary conditions) on the linking circle from bounding (anti-periodic) to non-bounding (periodic).

the same as ordinary toric code, given by the product of the Pauli $X$ operator on the six edges that meet the vertex. The plaquette term is modified compared to the ordinary toric code, and it is given by the product of $Z$ on the four edges surrounding the plaquette, as well as $X$ on the two edges that are perpendicular to the plaquette at two diagonal corners (the choice depends on the branching structure) and on two sides of the plaquette (see Appendix D for a review).

## 2.1 Electric and magnetic symmetries

The theory has the following (emergent) global symmetries, generated by invertible topological operators of various dimensions. We will organize them according to whether they carry magnetic flux (magnetic vs. electric operators) and the dimension of the subspace that supports the operator (a line, a surface or a domain wall). The electric operators supported on subspace $M$ of dimension $n$ are obtained by gauging the fermion parity of the fermionic invertible phase in $n$ spacetime dimension. The list of symmetries are as follows:

- $\mathbb{Z}_2$ 1-form symmetry generated by the codimension-two magnetic surface operator, around which the $\mathbb{Z}_2$ gauge field has nontrivial holonomy.[3]

- $\mathbb{Z}_2$ 2-form symmetry generated by the electric Wilson line of the $\mathbb{Z}_2$ gauge field. We can also view the line operator as gauging the fermion parity symmetry in the fermionic invertible phase in (0+1)D (see *e.g.* [44]).

- $\mathbb{Z}_2$ 1-form symmetry generated by the electric surface operator given by gauging the fermion parity symmetry in the Kitaev chain fermionic invertible phase in (1+1)D [68, 41, 44, 33].[4]

- $\mathbb{Z}_8$ 0-form symmetry generated by the electric domain wall operator given by gauging the fermion parity symmetry in the chiral p+ip fermionic invertible phase in (2+1)D [71]. This is related to the 16-fold way [72, 73] as we will explain below.

To sum up, the theory has $\mathbb{Z}_8$ 0-form symmetry, $\mathbb{Z}_2 \times \mathbb{Z}_2$ 1-form symmetry, and $\mathbb{Z}_2$ 2-form symmetry. We will specify how these symmetries interplay with one another in subsequent sections.

### 2.1.1 Reduced classification of symmetries: nontrivial symmetries in Hilbert space

The above symmetry groups generated by the electric operators can differ from the classification of fermionic invertible phases. In particular, (2+1)D fermionic invertible phases with $\mathbb{Z}_2^f$ fermion parity symmetry have a $\mathbb{Z}$ classification, generated by the $p+ip$ state, but in the above discussion, this gives rise to a $\mathbb{Z}_8$ 0-form symmetry of the (3+1)D $\mathbb{Z}_2$ gauge theory.

Moreover, since we can consider the subspaces that support the electric operators to be unorientable, it is appropriate to consider the classification of fermionic invertible phases with time-reversal or spatial reflection symmetry, which would naively give us a $\mathbb{Z}_8$ 1-form symmetry for the Kitaev chain [74]. However, this symmetry is actually reduced to $\mathbb{Z}_2$ and, as we will see, the symmetry generator squares to the electric Wilson line when it has support on a unorientable surface. This indicates a non-trivial mixing between the $\mathbb{Z}_2$ 1-form symmetry of the Kitaev chain defect, the $\mathbb{Z}_2$ 2-form symmetry of the Wilson line, and space-time orientation reversing defects.

The larger symmetry groups are extensions of the above minimal symmetries, but the extra symmetries act trivially on the Hilbert space, *i.e.* the code space, up to an overall phase, and so we do not consider them as symmetries any more than multiplication by complex numbers is a symmetry. In what follows we will explain in more detail the origin of these reductions of the symmetry. The details are technical and can be safely skipped for readers only interested in the application to logical gates.

**Reduction of the p+ip symmetry** As mentioned above, the symmetry generated by the electric domain wall operator decorated with p+ip is $\mathbb{Z}_8$ instead of the classification $\mathbb{Z}$ (where the domain walls are labelled by the half-integer chiral central charge $c_- = \nu/2$ for integer $\nu$).

Let us first explain the reduction of the $\mathbb{Z}$ symmetry to $\mathbb{Z}_{16}$. The domain wall with $\nu = 16$ differs from $\nu = 0$ by a bosonic $E_8$ phase (i.e. an invertible bosonic phase described by $(E_8)_1$ Chern-Simons theory). Thus the difference does not depend on the spin structure, which means that such defects do not interact with magnetic efects or electric defects, and thus act trivially on the ground state subspace up to an overall phase.

---

[3]Since the Wilson line is a fermion, on top of the magnetic defect the fermion does not have a well-defined spin structure.

[4]This surface operator is referred to as "electric" since it braids trivially with the Wilson line, in contrary to the "magnetic" surface operator which braids with the Wilson line by a sign.

Furthermore, there is a refinement that reduces the $\mathbb{Z}_{16}$ symmetry to $\mathbb{Z}_8$ symmetry: the effective action for the class $\nu = 8$ does not depend on the spin structure. To see this, we can start with the description of the class $\nu = 2$ as a $U(1)$ Chern-Simons theory [72]:

$$\frac{1}{4\pi} u du + \frac{\pi}{2\pi} u da \, , \tag{1}$$

where the local fermion is described by the Wilson line $e^{i \int u}$, and $a$ is the spin structure ($da = w_2$). The equation of motion for $u$ implies that the Wilson line $e^{i \oint u} = e^{\pi i \oint a}$ [75, 76].

When we change the spin structure by a $\mathbb{Z}_2$ background gauge field $B$, $a \to a + B$ (we take a lift to integer 1-cochain), the theory changes to

$$\frac{1}{4\pi} \int u du + \frac{\pi}{2\pi} \int u d(a + B) = \frac{1}{4\pi} \int u' du' + \frac{\pi}{2\pi} \int u' da - \frac{\pi}{4} \int B dB \, , \tag{2}$$

where $u' = u + \pi B$. Thus dependence of the theory on spin structure is captured by

$$(\nu = 2): \quad -\frac{\pi}{4} \int B dB \, . \tag{3}$$

The effective action of the $\nu = 8$ theory is 4 copies of $\nu = 2$, and it is independent of the spin structure:

$$4 \cdot \left( -\frac{\pi}{4} \int B dB \right) = 2\pi \int B \frac{dB}{2} = 0 \bmod 2\pi \, , \tag{4}$$

where we used $dB = 0 \bmod 2$. Thus we conclude that the effective action does not depend on the spin structure, although it requires a spin structure to be well-defined. This implies that the codimension-1 defect decorated with $\nu = 8$ does not interact with the magnetic defect at all, from which we conclude that it acts trivially on the Hilbert space, up to an overall phase. [5] This expectation is borne out by our concrete computations in later sections, where we see that all logical gates induced by sweeping the p+ip defect have order 8.

**Reduction of symmetry in terms of mixed gravitational anomaly**   The higher symmetries have a mixed gravitational anomaly that can be described by the (4+1)D topological term

$$\frac{2\pi}{16} \int \tilde{C}_1 \cup (p_1/3) + \pi \int C_3 \cup w_2 \, , \tag{5}$$

where $C_3$ is the background 3-form gauge field for the $\mathbb{Z}_2$ 2-form symmetry generated by the $\mathbb{Z}_2$ Wilson line of the emergent fermion, $\tilde{C}_1$ is a $\mathbb{Z}_{16}$ lift of the background 1-form gauge field $C_1$ for the $\mathbb{Z}_8$ symmetry, and $p_1$ is the first Pontryagin class. The first term describes the gravitational anomaly of the chiral p+ip state, while the second term describes the fact that the $\mathbb{Z}_2$ charge particle is a fermion. When the p+ip defect sweeps a closed 3-manifold $M^3$ Poincare dual to $\tilde{C}_1$, we obtain an overall phase obtained by evaluating $\frac{2\pi}{16} \int_{W^4} p_1/3$ on a 4-manifold $W^4$ such that $\partial W^4 = M^3$.

The reduction of the 0-form symmetry from $\mathbb{Z}$ to $\mathbb{Z}_8$ requires that the above anomaly action be invariant under a shift of $\tilde{C}_1 \to \tilde{C}_1 + 8\lambda_1$, with $\lambda_1$ an integer 1-cochain. This is indeed the case as long as we also shift $C_3 \to C_3 + w_2 \cup \lambda_1$. The effective action is invariant because of the property $p_1/3 = w_2^2 \bmod 2$ on orientable manifolds [78]. Such transformation means that the $\mathbb{Z}_8$ 0-form symmetry and the $\mathbb{Z}_2$ 2-form symmetry combine into a 3-group symmetry [28, 31].[6]

Across a domain wall decorated with 16 copies of the p+ip phase, $C_1 \to C_1 + d\phi$ where $\phi = 16$ on one side of the wall. The anomaly implies that on that side there is an extra term $2\pi \int \phi(p_1/3)$, which can be written as the

---

[5] One can also see the reduction to $\mathbb{Z}_8$ symmetry by the following alternative argument. In general, the dependence of the theory on the spin structure $a$ is encoded in the additional $\mathbb{Z}_2$ 0-form symmetry with the background gauge field $B$, since two different spin structures $a, a'$ are related by a background $\mathbb{Z}_2$ gauge field as $a' = a + B$. With this in mind, the spin structure dependence of the spin invertible phase with a partition function $Z(a)$ can be captured by the ratio of partition functions $Z'(a, B) := Z(a + B)Z^{-1}(a)$. The partition function $Z'(a, B)$ describes a spin invertible phase with the additional $\mathbb{Z}_2$ unitary internal symmetry, such that setting $B = 0$ leads to the trivial phase $Z'(a, 0) = 1$. The spin invertible phases with partition function $Z'$ in $D$ spacetime dimension are classified by $\Omega^D_{\mathrm{Spin}}(B\mathbb{Z}_2)/\Omega^D_{\mathrm{Spin}}(\mathrm{pt})$ [77]. In particular, setting $D = 3$ gives $\mathbb{Z}_8$, in agreement with the $\mathbb{Z}_8$ faithful symmetry found here.

[6] Another explanation for such correlated transformation is as follows. The $\nu = 8$ class in the 16-fold way [72] can be obtained from the $\nu = 0$ class by shifting the background of the symmetry generated by the emergent fermion [79]. The emergent fermion generates an 1-form symmetry on the wall, and if the background for the 1-form symmetry is shifted by $w_2$, the particles that transform under such 1-form symmetry will carry "additional projective representation" described by $w_2$, i.e. the spin of these particles will acquire additional fermion spin, such as changing from boson to a fermion and vice versa. The shift of the background can be described by $C_3 \to C_3 + w_2 \cup \lambda_1$, where $\lambda_1$ is the Poincaré dual for the wall with 8 copies of p+ip phase.

effective action $16\mathrm{CS}_{\mathrm{grav}}$ on the domain wall. This is the effective action of the bosonic $E_8$ phase, as expected from 16 copies of the p+ip phase.

Similarly, if we perform the transformation with $\phi = 8$ instead of 16, the first term in (5) produces an extra term $\pi \int (p_1/3)$ on the side with $\phi = 8$. However, there is a contribution from the second term in (5) under the accompanying transformation $C_3 \to C_3 + w_2 \cup d\phi/8$, which produces an extra term $\pi \int w_2 \cup w_2$ on the side of the domain wall with $\phi = 8$. Using $p_1/3 = w_2^2 \bmod 2$, we find the side with $\phi = 8$ has total term $2\pi \int p_1/3$ just as the previous case. Thus 8 copies of the domain wall also produces a bosonic domain wall.

We remark that the $\mathbb{Z}_{16}$ two-fold extension of the $\mathbb{Z}_8$ symmetry is obtained from another argument in [80, 81], but the faithful $\mathbb{Z}_8$ symmetry and the mixed anomaly with gravity are not discussed there.

**Reduction of the Kitaev chain symmetry, and gravitational anomaly**  The symmetry generated by the electric surface operator decorated with Kitaev chain is $\mathbb{Z}_2$ instead of the classification $\mathbb{Z}_8$ when the surface is unorientable. This can be seen as follows:

(1) The middle class in $\mathbb{Z}_8$ is bosonic, and thus the fourth power of the operator on an unorientable surface does not depend on spin structure and produces a decoupled trivial operator that does not act on the Hilbert space. There is a mixed gravitational anomaly corresponding to the (4+1)D topological term (see *e.g.* [40])

$$\frac{2\pi}{4} \int \tilde{C}_2 \cup W_3 + \pi \int C_3 \cup w_2 \ , \tag{6}$$

where $\tilde{C}_2$ is an integer cochain lift of the 2-form background $C_2$ for the $\mathbb{Z}_2$ 1-form symmetry, and $W_3 = d\tilde{w}_2/2$ is the integral third Stiefel-Whitney class ($\tilde{w}_2$ is an integer cochain lift of the second Stiefel-Whitney class). If we change the lift $\tilde{C}_2 \to \tilde{C}_2 + 2\tilde{\lambda}_2$ for some integer cochain lift of $\mathbb{Z}_2$ 2-cocycle $\lambda_2$, the first term changes by $\pi \int \lambda_2 \cup W_3 = \pi \int \frac{d\tilde{\lambda}_2}{2} \cup w_2$. Thus the change can be compensated by the transformation $C_3 \to C_3 + \frac{d\tilde{\lambda}_2}{2}$. Such a transformation indicates that the 1-form symmetry and the 2-form symmetry combine into a 3-group.

Under a transformation $\tilde{C}_2 \to \tilde{C}_2 + 4\tilde{\lambda}$ where $\tilde{\lambda}$ is a lift of the $\mathbb{Z}_2$ 2-cocycle $\lambda$, the anomaly inflow produces a boundary term $\pi \int \lambda \cup w_2 = \pi \int_{\mathrm{PD}(\lambda)} w_2 = \pi \int_{\mathrm{PD}(\lambda)} w_1^2$, where PD denotes the Poincaré dual. This is the fourth power of the operator, and the operator does not depend on the spin structure.

(2) The square of the operator on an unorientable surface reduces to the electric Wilson line operator on the Poincaré dual of $w_1$ (See Appendix A). Let us show this using the effective field theory description of Kitaev chain given by the $\mathbb{Z}_2$ gauge theory with the action [82, 44, 83, 36]

$$\frac{2\pi}{8} \mathrm{ABK}(\Sigma, a) \ , \tag{7}$$

where $\Sigma$ is a closed 2d surface that supports a Kitaev chain (which may be unorientable), and where $a$ is the induced pin$^-$ structure of the surface $\Sigma$ (and we identify it with the bulk gauge field). $\mathrm{ABK}(\Sigma, a)$ denotes the Arf-Brown-Kervaire (ABK) invariant of the pin$^-$ surface valued in $\mathbb{Z}_8$, which indicates the $\mathbb{Z}_8$ classification of the Kitaev chain in the presence of the time reversal or reflection symmetry. One can express the ABK invariant in terms of a $\mathbb{Z}_2$ gauge theory with support at $\Sigma$,

$$\frac{\pi}{2} \int_{\Sigma} q_a(b) \ , \tag{8}$$

where $q$ is the quadratic function that refines the intersection between $\mathbb{Z}_2$-valued 1-cocycles on the surface, and $b$ is a dynamical ordinary $\mathbb{Z}_2$ gauge field with bosonic Wilson line, which is defined on the surface $\Sigma$ (instead of the whole 4D spacetime). Summing over configurations of the dynamical gauge field $b$ yields the ABK invariant evaluated at the surface $\Sigma$.

Taking two copies of the theory (denote the dynamical gauge fields by $b, b'$ in the two copies) gives the total effective action:

$$\frac{\pi}{2} \int q_a(b) + \frac{\pi}{2} \int q_a(b') = \frac{\pi}{2} \int q_a(\tilde{b}) + \pi \int \tilde{b} \cup b + \pi \int b \cup b = \frac{\pi}{2} \int q_a(\tilde{b}) + \pi \int (\tilde{b} + w_1) \cup b \ , \tag{9}$$

where $\tilde{b} = b + b'$, the first equality uses the property $q(x + y) = q(x) + q(y) + 2x \cup y \bmod 4$ for any $\mathbb{Z}_2$-valued 1-cocycles $x, y$ [82, 44, 36], and the second equality uses $b \cup b = w_1 \cup b$ on the surface, where $w_1$ denotes the

1st Stiefel-Whitney class on the surface $\Sigma$. Integrating out $b$ imposes $\tilde{b} = w_1(\Sigma)$, and thus two copies of the Kitaev chain phase can be described by the effective action

$$\frac{\pi}{2} \int_\Sigma q_a(w_1(\Sigma)) \ . \tag{10}$$

Let us examine how the effective action depends on $a$: if we write $a = a_0 + s$ for some fixed reference pin$^-$ structure $a_0$, then the effective action reduces to $\pi \int w_1 \cup s$ up to terms independent of $s$. In other words, the square of the electric surface operator decorated with the Kitaev chain phase produces an electric Wilson line of $a$ on the loop that reverses the orientation of the surface.

### 2.1.2  Symmetry generator with non-Abelian topological order can be invertible

We note that gauging the fermion parity symmetry in p+ip topological superconductor produces a non-Abelian topological order [72], but the domain wall decorated with such topological order nevertheless generates an invertible symmetry. Let us explain how the two properties coexist.

The closed magnetic defects are invertible, but once they end on the p+ip domain wall they give rise to non-Abelian anyons on the domain wall. The reason is that invertible defects such as magnetic defects of $\mathbb{Z}_2$ gauge theory can become non-invertible once they have a boundary (such as ending on the gauged SPT defect) which can carry additional degrees of freedom. In particular, the magnetic defect ending on the p+ip domain wall with additional Majorana zero mode (see e.g. the lattice model for p+ip phase discussed in [72]).

Such examples are ubiquitous. For instance, the $\mathbb{Z}_2$ electromagnetic duality in (2+1)D $\mathbb{Z}_2$ gauge theory can end on the boundary to become the non-invertible Kramers-Wannier duality defect [16, 84, 85], and similar examples in higher dimensions discussed in *e.g.* [86, 87].

## 2.2  $\mathbb{Z}_8$ p+ip 0-form symmetry defines automorphism of 1-form symmetry

We will show that the $\mathbb{Z}_8$ ordinary symmetry generated by the domain wall decorated with p+ip fermionic invertible phase (we will refer to it as the "p+ip symmetry") generates an automorphism transformation on the $\mathbb{Z}_2 \times \mathbb{Z}_2$ 1-form symmetry for the magnetic flux surface operator $V^{(2)}$ and the electric surface operator $U^{(2)}$ decorated with Kitaev chain fermionic invertible phase. Conjugating by the p+ip domain wall operator generates the following automorphism transformation:

$$U^{(2)} \to U^{(2)}, \quad V^{(2)} \to V^{(2)}U^{(2)} \ . \tag{11}$$

The above automorphism can be derived by the charge-flux attachment argument [40]. Consider a magnetic flux defect intersecting the domain wall decorated with p+ip. The intersection is a magnetic vortex defect on the p+ip domain wall, and such vortex has Majorana zero mode [72]. Such Majorana zero mode is the boundary mode of the electric Kitaev chain surface defect, and thus the magnetic flux surface defect intersecting the domain wall is transformed into the "dyonic" defect with additional electric Kitaev chain surface defect, as in (11).

We remark that the charge-flux attachment also follows from twisted dimensional reduction of the fermionic invertible phases on a circle with periodic boundary condition, similar to the reduction of bosonic SPT phases on a circle with nontrivial holonomy as discussed in [40]. The p+ip phase can be described by a single free Majorana fermion field theory in (2+1)D with a negative mass (where we choose a regulator such that fermion with positive mass belongs to the trivial phase). Reducing the fermion theory on a circle by demanding the Majorana field to be independent of the circle direction (which requires the periodic boundary condition on the circle, as in the presence of magnetic defect) produces the theory of a single Majorana fermion with a negative mass in (1+1)D, and it is the effective field theory for the Kitaev chain phase. The argument generalizes to any dimensions. An example of twisted reduction of (1+1)D Kitaev chain phase to (0+1)D fermionic invertible phase is discussed in [88].

The automorphism is also obtained from another argument in [80].

We can describe such automorphism using the background gauge fields for the corresponding symmetries generated by the defects. The background gauge fields satisfy

$$dC_2 = B_2 \cup C_1. \tag{12}$$

where $C_1$ is the background gauge field for the $\mathbb{Z}_8$ 0-form symmetry, and $C_2, B_2$ are the background gauge fields for the $\mathbb{Z}_2 \times \mathbb{Z}_2$ 1-form symmetry generated by the electric Kitaev surface operator and the magnetic flux surface operator.

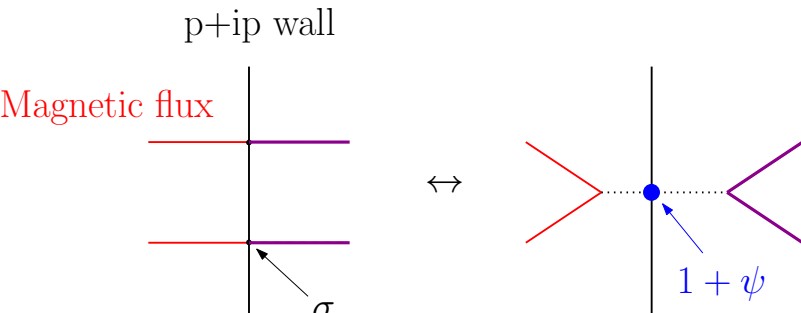

Figure 1: Two fusion configurations for fusing the p+ip domain wall with (1) a pair of magnetic flux surface operators indicated by the red lines entering the wall (black line), and (2) a pair of "dyonic" surface operators indicated by the purple lines that carry magnetic flux and are decorated with the Kitaev chain phase due to the automorphism (11). The dotted line in the configuration on the right is the trivial defect. One of the configurations has additional $1 + \psi$ defect due to the fusion rule $\sigma \times \sigma = 1 + \psi$ for vortex $\sigma$ in p+ip phase [72]. We can relate the two configurations by the "defect-valued associator" whose value is given by the non-invertible defect $1 + \psi$. Such a (generalized) associator is similar to the 2-group symmetry, which can be regarded as an "F symbol" that takes value in abelian anyons for fusing the domain walls that generate 0-form symmetry [17, 31].

## 2.3    Non-invertible operator-valued associator

The electric Wilson line on the p+ip domain wall is the fermion $\psi$. It obeys the following Ising fusion rule with the magnetic vortex $\sigma$: $\sigma \times \sigma = 1 + \psi$, $\sigma \times \psi = \sigma$ [72].

The first fusion rule $\sigma \times \sigma = 1 + \psi$ implies that if there are two magnetic flux surface defects intersect the p+ip domain wall, and we bring the intersection locus together, there will be the condensation $1 + \psi$ left behind at the intersection locus. This also follows from the fusion of two open Kitaev chain defects. The extra condensation defect can be viewed as an operator-valued associator for fusing magnetic flux surfaces and the p+ip domain walls as shown in Figure 1, similar to the 2-group symmetry as operator-valued associator for domain walls [31].

Similarly, the fusion rule $\sigma \times \psi = \sigma$ implies that the intersection locus of a single magnetic flux with the domain wall can absorb an electric Wilson line.

## 2.4    $\mathbb{Z}_4 \subset \mathbb{Z}_8$ 0-form symmetry and 3-group

Let us consider the $\mathbb{Z}_4 \subset \mathbb{Z}_8$ symmetry, i.e. the even classes in the 16-fold way [72], indexed by $\nu = 2j \mod 16$ for $j$ integer. The topological order on the domain wall associated with such symmetries is Abelian. In particular, since the vortex on such domain wall is Abelian (it obeys the fusion $m \times m = \psi$ when $\nu = 4k + 2$ and $m \times m = 1$ when $\nu = 4k$, for $k$ integer) [72], the associator for the domain wall and the magnetic flux is also invertible.

In the following we will discuss in detail the 3-group structure that arises from the interplay between such $\mathbb{Z}_4$ 0-form symmetry generated by the domain wall with $\nu = 2$ in the 16-fold way (which we sometimes refer to as the Chern insulator), the $\mathbb{Z}_2 \times \mathbb{Z}_2$ 1-form symmetry generated by the magnetic flux surface and the surface decorated with the Kitaev chain invertible phase, and the $\mathbb{Z}_2$ 2-form symmetry generated by the Wilson line of the $\mathbb{Z}_2$ gauge theory. It turns out that the 3-group structure for the symmetries translates into the commutation relations obeyed by the corresponding symmetry generators, allowing us to obtain logical Clifford gates implemented by the $\mathbb{Z}_4$ 0-form symmetry action.

### 2.4.1    3-group symmetry structure

The symmetries form a 3-group structure, where three stands for the highest codimension of the symmetry defects (the Wilson line). The $\nu = 2$ generator of the $\mathbb{Z}_4$ 0-form symmetry does not permute the 1-form symmetry. The higher-group structure arises from the different junctions of the domain walls and the surfaces that can produce the Wilson line.

The 3-group symmetry structure between the 2-form symmetry and the 1-form symmetries, i.e. the junctions of surface defects that can produce the Wilson line, is discussed in [89, 33, 40]. Here we will focus on the three group structure from the junctions that involve the domain walls (or both the domain walls and the surface defects). There are two junctions that can produce Wilson line:

**Junctions of domain wall and surface defects that produces Wilson line**   The first junction consists of intersection of the domain wall with the tri-junction for the magnetic flux surface defects. We note that such tri-junction can be described by background $B_2$ with nontrivial $d\tilde{B}_2/2$. On the $\nu = 2$ domain wall, the magnetic flux $m$ obey the fusion rule $m \times m = \psi$ [72]. Thus then the junction of fusing two magnetic fluxes intersect with the domain wall, there is additional Wilson line $\psi$. See Figure 2 for an illustration. Such junction thus contributes to the 3-group symmetry structure, and was also studied in other examples in [40].

The second junction consists of the intersection of the magnetic flux surface defect with the tri-junction of domain wall that fuses two $\nu = 4$ domain walls into $\nu = 8$ domain wall. We note that such tri-junction can be described by the background $C_1$ with nontrivial $d\tilde{C}_1/4$. This can be understood from the fusion of two semion magnetic fluxes on the two $\nu = 4$ domain walls into a fermion (and the 3 fermions in the $\nu = 8$ phase are all on equal footing) Another way to understand such property is from the dependence of spin structure for the $\nu = 4$ domain wall, which can be captured by twice of (3), and it is the effective action of the Levin-Gu phase. The junction is thus essentially discussed in [32, 40] (see Figure 3), and when the magnetic flux intersect the junction there is additional electric charge.

**3-group symmetry structure as background fields relation**   We can also describe the 3-group symmetry structure in terms of equations satisfied for closed (flat) background gauge field configurations for the higher symmetries. Let us introduce the following background gauge fields for the symmetries:

- Background 2-form gauge field $B_2$ for the $\mathbb{Z}_2$ 1-form symmetry generated by the magnetic flux surface defect.

- Background 2-form gauge field $C_2$ for the $\mathbb{Z}_2$ 1-form symmetry generated by the electric surface defect decorated with the Kitaev chain invertible phase.

- Background 3-form gauge field $C_3$ for the $\mathbb{Z}_2$ 2-form symmetry generated by the electric Wilson line.

- Background 1-form gauge field $C_1$ for the $\mathbb{Z}_4$ subgroup 0-form symmetry generated by the electric domain wall defect decorated with the Chern insulator (i.e., $\nu = 2$ phase which carries $c_- = 1$).

The 3-group symmetry structure can be described by the following relation among the background fields:

$$dC_3 = Sq^2(C_2) + B_2 \cup C_2 + \left( \frac{d\tilde{B}_2}{2} + w_3 \right) \cup C_1 + (B_2 + w_2) \cup \frac{d\tilde{C}_1}{4} \ , \tag{13}$$

where the first two terms on the right hand side are discussed in [89, 40] and describe the fact that the intersection of the Kitaev string and magnetic string nucleates a fermion.

The last two terms on the right hand side are a new contribution to the 3-group structure and corresponds to the two junctions discussed above. In the last two terms, $\tilde{B}_2$ is a lift of the $\mathbb{Z}_2$ 2-form gauge field to $\mathbb{Z}_4$ value, and $\tilde{C}_1$ is a lift of $\mathbb{Z}_4$ 1-form gauge field to $\mathbb{Z}_8$ value. The relation between the background fields does not depend on the choice of lift.[7] The three-cocycle $d\tilde{B}_2/2 + w_3$ is only nontrivial in the presence of the junction of fusing two magnetic fluxes, and $(d\tilde{B}_2/2 + w_3) \cup C_1$ is nontrivial only when the junction intersects the domain wall that generates the subgroup 0-form symmetry. This is the junction that produces the electric Wilson line defect discussed above, and it gives nontrivial background $C_3$.

**3-group symmetry in effective field theory**   We can also derive the above equations for the closed background gauge fields of the 3-group symmetry explicitly in the following way. We will focus on the 1-form symmetry generated by the magnetic surface operator and the $\mathbb{Z}_4$ subgroup 0-form symmetry. The 0-form symmetry is generated by the domain wall decorated with the gravitational Chern-Simons term $-2\text{CS}_{\text{grav}}$. Let us examine how the theory depends on the spin structure $a$. The gravitational Chern-Simons term $2\text{CS}_{\text{grav}}$ can be expressed in terms of an emergent $U(1)$ gauge field $u$ as follows: (see *e.g.* [75, 90])

$$-2\text{CS}_{\text{grav}}[a] \quad \longleftrightarrow \quad \frac{1}{4\pi} u du + \pi \frac{du}{2\pi} a = \frac{1}{4\pi} u' du' - \frac{\pi}{4} a da \ , \tag{14}$$

where the left hand side includes $[a]$ to remind ourselves the theory depends on the spin structure $a$; on the right hand side we use the change of variables $u' = u + \pi a$, and we take a lift of $a$ to be an integer 1-cochain with holonomy

---

[7]The object $d\tilde{B}_2/2$ is the image of $B_2$ under the connecting homomorphism for the long exact sequence in cohomology from the group extension short exact sequence $1 \to \mathbb{Z}_2 \to \mathbb{Z}_4 \to \mathbb{Z}_2 \to 1$. Similarly, the object $d\tilde{C}_1/4$ is the image of $C_1$ under the connecting homomorphism for the short exact sequence $1 \to \mathbb{Z}_2 \to \mathbb{Z}_8 \to \mathbb{Z}_4 \to 1$.

$0, 1$. The equation of motion for $u$ relates the holonomy $e^{i \int u} = \pm 1$ with the spin structure. From the last expression, the theory on the wall depends on the spin structure by

$$- \int \frac{\pi}{4} a \cup da .$$ (15)

In the presence of background $C_1$, the action on the spacetime is modified with the term

$$-\frac{\pi}{4} \int a \cup da \cup \tilde{C}_1 .$$ (16)

where $\tilde{C}_1$ is a lift of the background $C_1$ to $\mathbb{Z}_8$ value.

If we turn on background $B_2$ for the 1-form symmetry generated by the magentic flux surface operator, the quantization of the flux is modified to be $da = w_2 + B_2$. The action (16) needs additional corrections that depend on $B_2$ in order to be invariant under the transformation $a \to a + 2\tilde{\lambda}_1$, where $\tilde{\lambda}_1$ is an integer lift of a $\mathbb{Z}_2$ 1-cocycle $\lambda_1$. The transformation changes the action by

$$\pi \int \lambda_1 \cup da \cup C_1 = \pi \int \lambda_1 \cup (B_2 + w_2) \cup C_1 .$$ (17)

Thus the correction is

$$\frac{\pi}{2} \int a \cup (\tilde{B}_2 + \tilde{w}_2) \cup C_1 ,$$ (18)

where $\tilde{B}_2, \tilde{w}_2$ are lifts of $B_2, w_2$ from $\mathbb{Z}_2$ value to $\mathbb{Z}_4$ value.

However, the theory with the correction term by itself is not well-defined: under a gauge transformation $a \to a + d\phi$, the action is not invariant due to both the original action and the correction term. The original action (16) transforms under $a \to a + d\phi$ by

$$-\frac{\pi}{4} \int d\phi da \tilde{C}_1 = \pi \int \phi da \cup \frac{d\tilde{C}_1}{4} = \pi \int \phi (B_2 + w_2) \cup \frac{d\tilde{C}_1}{4} .$$ (19)

The correction (18) transforms by

$$\pi \int \phi \cup \frac{d}{2} \left( (\tilde{B}_2 + \tilde{w}_2) \cup \tilde{C}_1 \right) .$$ (20)

We can compensate the transformation $\phi$ by including nontrivial background field $C_3$ for the 2-form symmetry generated by the Wilson line, which couples to the theory as

$$\pi \int a \cup C_3 .$$ (21)

The total couplings are invariant under the transformation $a \to a + d\phi$ provided the background fields satisfy

$$dC_3 = \frac{d}{2} \left( \left( \tilde{B}_2 + \tilde{w}_2 \right) \cup C_1 \right) + (B_2 + w_2) \cup \frac{d\tilde{C}_1}{4} = \left( \frac{d\tilde{B}_2}{2} + w_3 \right) \cup C_1 + (B_2 + w_2) \cup \frac{d\tilde{C}_1}{4} \mod 2 ,$$ (22)

where we used the property that $dC_1$ is a multiple of 8. This reproduces (13) with $C_2 = 0$, and $w_3 = d\tilde{w}_2/2 \mod 2$ on orientable manifolds.

We note that although $C_1$ is the background for $\mathbb{Z}_4 \subset \mathbb{Z}_8$ subgroup 0-form symmetry, the coupling (16) is not invariant under changing the $\mathbb{Z}_8$ value lift $\tilde{C}_1$: if we change the lift $\tilde{C}_1 \to \tilde{C}_1 + 4\lambda_1$ with an integer 1-cochain $\lambda_1$, the coupling changes by

$$\pi \int a \cup da \cup \lambda_1 = \pi \int a \cup (B_2 + w_2) \cup \lambda_1 .$$ (23)

Thus for the coupling to be well-defined, the background $C_3$ must also transform as $C_3 \to C_3 + w_2 \cup \lambda_1$. This is consistent with the relation (22).

We note that the first term on the right hand side in (22) can also be understood from the dependence on the spin$^c$ structure on the domain wall: if we turn on a spin$^c$ connection $A$, the domain wall theory (16) becomes $-\frac{1}{4\pi} udu + \frac{1}{2\pi} udA$, and the dependence on $A$ is the same as the insertion of the local fermion particle (*i.e.* the Wilson line $e^{i \int u}$) at the Poincaré dual of $dA/2\pi$ with respect to the domain wall. This is consistent with the transformation $w_3 \to w_3 + d\lambda_2$, $C_3 \to C_3 + \lambda_2 \cup C_1$, where for global transformation $\lambda_2 = dA/2\pi \mod 2$.

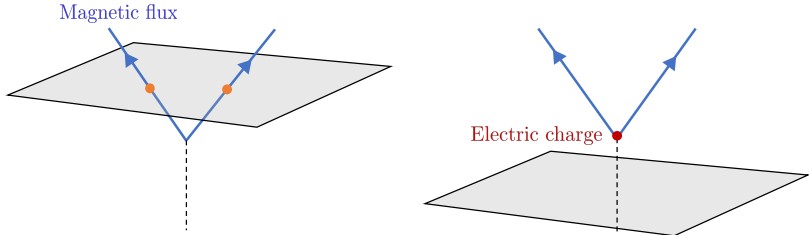

Figure 2: A pair of magnetic vortices (orange dots) are placed at the codimension-1 domain wall defect (gray sheet) that generates subgroup $\mathbb{Z}_4$ 0-form symmetry carrying the $\nu = 2$ phase. When two magnetic fluxes fuse on the domain wall, it produces the Wilson line (red dot) from the fusion rules in the class two theory $U(1)_4$ in the 16-fold way [72].

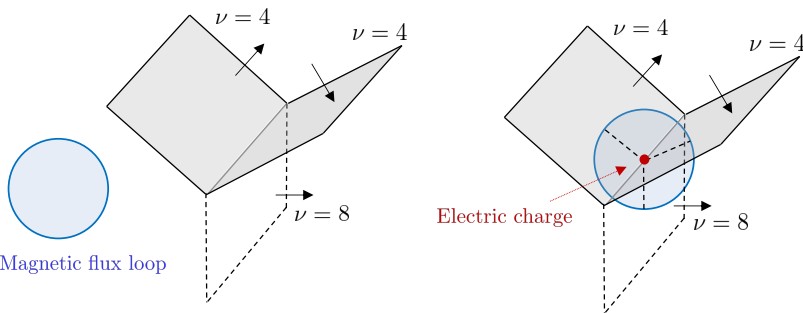

Figure 3: A pair of $\mathbb{Z}_2$ 0-form symmetry defects carrying the $\nu = 4$ phase fuses into a trivial symmetry defect carrying the $\nu = 8$ phase. The arrow indicates the orientation of the defect. The magnetic flux at each of the $\nu = 4$ symmetry defects behaves as a semion. The two semions fuse into a fermionic particle at the junction, which is identified as an electric particle. This is one symptom of the 3-group structure of the symmetry.

# 3 Logical gates in (3+1)D $\mathbb{Z}_2$ gauge theory with emergent fermion

In this section, we derive the action of the logical gates of (3+1)D $\mathbb{Z}_2$ toric code with an emergent fermion, which are implemented by the emergent global symmetry of $\mathbb{Z}_2$ gauge theory discussed above. We consider the $(3+1)$D $\mathbb{Z}_2$ gauge theory with an emergent fermion, and derive the action of the emergent symmetry on the Hilbert space. The 3d space is taken to be a generic oriented 3-manifold $M^3$. Recall that the (3+1)D $\mathbb{Z}_2$ gauge theory has the following global symmetries, all of which give rise to the logical gates of the (3+1)D toric code with an emergent fermion.

- $\mathbb{Z}_8$ 0-form symmetry; the generator is given by decorating the codimension-1 defect with the (2+1)D p+ip superconductor with $c_- = 1/2$.

- $\mathbb{Z}_2$ 1-form symmetry generated by the magnetic surface operator of codimension-2.

- $\mathbb{Z}_2$ 1-form symmetry generated by decorating the codimension-2 defect with the (1+1)D spin invertible phase given by Kitaev's Majorana chain.

- $\mathbb{Z}_2$ 2-form symmetry generated by the electric Wilson line operator of an emergent fermion.

First of all, when we have $|H_1(M^3, \mathbb{Z}_2)| = 2^N$ there are $N$ logical qubits. The Pauli $X_j$ operator for the $j$-th qubit with $1 \le j \le N$ is implemented by the magnetic surface operator on a closed surface $\Sigma_j$, where the set of surfaces $\{\Sigma_j\}$ spans the basis of the 2nd homology $H_2(M^3, \mathbb{Z}_2)$.

The Pauli $Z_j$ operator is then encoded by an electric Wilson line operator along the curve $\gamma_j$, which is dual to $\Sigma_j$ by the intersection pairing satisfying $\#(\gamma_j, \Sigma_k) = \delta_{jk} \pmod 2$.

## 3.1 Logical gate from Kitaev chain defect

Let us denote the dynamical $\mathbb{Z}_2$ gauge field for the (3+1)D $\mathbb{Z}_2$ gauge theory on a 3d space as $a \in C^1(M^3, \mathbb{Z}_2)$ satisfying $da = w_2(M^3)$, which represents a dynamical spin structure of the 3d space. Each eigenstate of the logical Pauli $\{Z_j\}$ operator can then be represented by a state $|\{a\}\rangle$, which is the state obtained by equal weight superposition over all

gauge equivalent configurations of the $\mathbb{Z}_2$ gauge field $a$. Each state $|\{a\}\rangle$ corresponds to a choice of spin structure on $M^3$. The $Z_j$ eigenvalue can be read by evaluating the holonomy $(-1)^{\int a}$ along the curve $\gamma_j$.

Let us now derive the action of the 1-form symmetry operator $\mathcal{W}_{\mathrm{K}}(\Sigma_j)$ corresponding to the Kitaev chain defect. We take $\{\Sigma_j\}$, for $j = 1, \cdots, \dim H_2(M^2, \mathbb{Z}_2)$ to define a representative basis for the second homology group $H_2(M^3, \mathbb{Z}_2)$, where $\Sigma_j$ can be unorientable.

The surface operator $\mathcal{W}_{\mathrm{K}}(\Sigma_j)$ then evaluates the Arf-Brown-Kervaire (ABK) invariant of a pin$^-$ surface $\Sigma_j$,

$$\mathcal{W}_{\mathrm{K}}(\Sigma_j) \, |\{a\}\rangle = \exp\left(\frac{2\pi i}{8} \mathrm{ABK}(\Sigma_j, a)\right) |\{a\}\rangle, \tag{24}$$

Here, the ABK invariant evaluates the partition function of the Kitaev chain on the (1+1)D pin$^-$ surface [44, 83, 36]. The pin$^-$ structure on $\Sigma_j$ is induced by the spin structure of a 3-manifold $M^3$, where the spin structure of $M^3$ along $\gamma_j$ is specified by the configuration of the $\mathbb{Z}_2$ gauge field $a$. The above action of the logical gate can be expressed by the following Clifford unitary (see Appendix A for derivations):

$$\mathcal{W}_{\mathrm{K}}(\Sigma_j) = \prod_{\substack{k,l \\ k<l, j\neq k, j\neq l}} CZ_{k,l}^{\int_{\Sigma_j} \sigma_k \sigma_l} \cdot \prod_{\substack{k \\ j\neq k}} CZ_{j,k}^{\int_{\Sigma_j} \sigma_k \sigma_k} (S_k^\dagger)^{\int_{\Sigma_j} \sigma_k \sigma_k} \cdot (e^{\frac{2\pi i}{8}} S_j^\dagger)^{\#(\Sigma_j, \Sigma_j, \Sigma_j)}$$

$$= \prod_{\substack{k,l \\ k<l, j\neq k, j\neq l}} CZ_{k,l}^{\int_{M^3} \sigma_j \sigma_k \sigma_l} \cdot \prod_{\substack{k \\ j\neq k}} CZ_{j,k}^{\int_{M^3} \sigma_j \sigma_k \sigma_k} (S_k^\dagger)^{\int_{M^3} \sigma_j \sigma_k \sigma_k} \cdot (e^{\frac{2\pi i}{8}} S_j^\dagger)^{\int_{M^3} \sigma_j \sigma_j \sigma_j}, \tag{25}$$

where $\sigma_j \in H^1(M^3, \mathbb{Z}_2)$ is the Poincaré dual of the surface $\Sigma_j$. The triple intersection is defined to be $\#(\Sigma_j, \Sigma_j, \Sigma_j) = \int_{M^3} \sigma_j \sigma_j \sigma_j$.

## 3.2 Logical gate from p+ip defect

Now we want to derive the action of the 0-form symmetry arising from the p+ip superconductor. For this purpose, it is convenient to first consider the 0-form symmetry of (3+1)D $\mathbb{Z}_2 \times \mathbb{Z}_2$ gauge theory with two emergent fermions. This theory has a 0-form $\mathbb{Z}_8$ symmetry generated by a decoration of (p+ip)×(p−ip) state on the codimension-1 defect. This state is regarded as (2+1)D $\mathbb{Z}_2 \times \mathbb{Z}_2^f$ SPT phase with the $\mathbb{Z}_8$ classification. Our strategy is to first derive the action of this (p+ip)×(p−ip) defect on the $\mathbb{Z}_2 \times \mathbb{Z}_2$ gauge theory in the form of the tensor product operator $V_{\mathrm{f}} \otimes V_{\mathrm{f}'}$, where each $V_{\mathrm{f}}, V_{\mathrm{f}'}$ acts on each Hilbert space of $\mathbb{Z}_2$ gauge theory with a single fermion. The action of the p+ip defect can then be obtained by $V_{\mathrm{f}}$.

The details of the computation is relegated to Appendix B. The expression of the p+ip logical gate is given as follows (up to overall phase):

$$V_{\mathrm{p+ip}}(M^3) = \prod_{\substack{k,l \\ j<k<l}} (CCZ_{j,k,l})^{\int_{M^3} \sigma_j \sigma_k \sigma_l} \cdot \prod_{\substack{k \\ j<k}} (CS_{j,k}^\dagger)^{\int_{M^3} \sigma_j \sigma_k \sigma_k} \cdot \prod_j (T_j)^{\int_{M^3} \sigma_j \sigma_j \sigma_j} . \tag{26}$$

This p+ip logical gate indeed gives the order 8 operation, as its 8th power becomes the overall phase acting trivially on the Hilbert space. This is consistent with our discussions in Sec. 2.1 showing that the 0-form symmetry reduces to $\mathbb{Z}_8$.

In the following sections, we will describe several examples of logical gates from the p+ip defects on different spatial 3-manifolds: for $M^3 = T^3, T^2 \rtimes_\mathcal{C} S^1, \mathbb{RP}^3$, where $T^2 \rtimes_\mathcal{C} S^1$ denotes a mapping torus of $T^2$ twisted by the modular $\mathcal{S}^2 = \mathcal{C}$ transformation. For each choice of a spatial 3-manifold $M^3$, the p+ip logical gate produces $CCZ$, Controlled-$S^\dagger$ and $T$ gate, respectively.

# 4 Example: review of fault-tolerant CCZ gate in (3+1)D $\mathbb{Z}_2$ fermionic toric code

In this section, we review the microscopic realization for the the p+ip logical gate on a torus $M^3 = T^3$, which has been studied in [69]. We will see that our formula for the p+ip logical gate Eq. (26) on $M^3 = T^3$ reproduces the results in [69].

## 4.1 Pumping a Chern insulator through a 3d torus

We first recall the construction of a unitary $U_{\text{Chern}}$ acting on a 3d fermionic system on a cubic lattice, which is regarded as pumping the Chern insulator through the whole 3d space [69]. This unitary turns out to be an exact 0-form symmetry of a trivial atomic insulator state, so generates the 0-form symmetry of the (3+1)D fermionic toric code after gauging the $\mathbb{Z}_2^f$ symmetry. This 0-form symmetry corresponds to the symmetry defect obtained by a decoration with the Chern insulator with integer $c_-$ studied in Sec. 2.4.

To construct this unitary, recall that the Chern insulator can be realized with a 2d free fermion Hamiltonian with two energy bands:

$$H_{\text{Chern}}(\vec{k}_{\text{2d}}) = c^X(\vec{k}_{\text{2d}})X^{\text{fl}} + c^Y(\vec{k}_{\text{2d}})Y^{\text{fl}} + c^Z(\vec{k}_{\text{2d}})Z^{\text{fl}} \tag{27}$$

where the Pauli matrices for the flavor index are denoted as $X^{\text{fl}}, Y^{\text{fl}}, Z^{\text{fl}}$. The real coefficient $\{c^X, c^Y, c^Z\}$ satisfies $(c^X)^2 + (c^Y)^2 + (c^Z)^2 = 1$, and it is a map $T^2 \to S^2$ with non-trivial winding number 1. An explicit choice of the function $\{c^X, c^Y, c^Z\}$ is made in Appendix C.

The inverse phase of the Chern insulator carrying the winding number $-1$ can be obtained by flipping the sign of the second term:

$$\overline{H}_{\text{Chern}}(\vec{k}_{\text{2d}}) = c^X(\vec{k}_{\text{2d}})X^{\text{fl}} - c^Y(\vec{k}_{\text{2d}})Y^{\text{fl}} + c^Z(\vec{k}_{\text{2d}})Z^{\text{fl}} \tag{28}$$

By stacking the above two theories $H_{\text{Chern}}$ and $\overline{H}_{\text{Chern}}$ we obtain the paired Hamiltonian with four energy bands

$$H_{\text{pair}}(\vec{k}_{\text{2d}}) = c^X(\vec{k}_{\text{2d}})X^{\text{fl}} + c^Y(\vec{k}_{\text{2d}})Z^{\text{layer}}Y^{\text{fl}} + c^Z(\vec{k}_{\text{2d}})Z^{\text{fl}} \tag{29}$$

where the Pauli matrices for the layer index is denoted as $X^{\text{layer}}, Y^{\text{layer}}, Z^{\text{layer}}$. A trivial atomic insulator can be transformed into this paired Hamiltonian $H_{\text{pair}}$ by a unitary $U^{\text{nucl}}$ obtained in [69] (see also Appendix C for an explicit definition of $U^{\text{nucl}}$)

$$U^{\text{nucl}\dagger} H_{\text{pair}} U^{\text{nucl}} = Z^{\text{fl}} \tag{30}$$

In other words, this unitary can nucleate a pair of topological insulators with the Chern number $\pm 1$ starting with a pair of trivial atomic insulators.

One can now construct a unitary pumping the Chern insulator through the 3d cubic lattice on a torus $T^3$ as follows; we take the 3d system to be the $2N$ layers of 2d square lattices stacked along the $z$ direction. Starting with the $2N$ layers of the trivial atomic insulators, we first nucleate the pair of the Chern insulator and its opposite on two layers $(2j-1, 2j)$ by the unitary $U^{\text{nucl}}$. Next, we annihilate the pair of the nontrivial insulators into the trivial one by applying the unitary $U^{\text{annih}} := X^{\text{layer}}U^{\text{nucl}\dagger}X^{\text{layer}}$ at the pair of layers $(2j, 2j+1)$ (see Figure 4). The whole process is summarized in the following single unitary

$$U_{\text{Chern}} = \prod_{\vec{k}_{\text{2d}}} U_{\text{Chern}}(\vec{k}_{\text{2d}}) \, , \tag{31}$$

with

$$U_{\text{Chern}}(\vec{k}_{\text{2d}}) = \prod_{1 \le j \le N} U^{\text{annih}}_{2j, 2j+1}(\vec{k}_{\text{2d}}) \prod_{1 \le j \le N} U^{\text{nucl}}_{2j-1, 2j}(\vec{k}_{\text{2d}}). \tag{32}$$

## 4.2 Pumping a p+ip superconductor through a 3d torus

Since the above unitary is symmetric under the translation by two layers in the $z$ direction, one can Fourier transform in the $z$ direction, which brings $U_{\text{Chern}}$ into a $4 \times 4$ matrix for the bilayer $U_{\text{Chern}}(\vec{k}_{\text{2d}}, k_z)$. This operator $U_{\text{Chern}}(\vec{k}_{\text{2d}}, k_z)$ is shown to commute with $Z^{\text{fl}}$, $U_{\text{Chern}}(\vec{k}_{\text{2d}}, k_z)$ is expressed in the block diagonal form for each eigenspace of $Z^{\text{fl}} = (-1)^\alpha$,

$$U_{\text{Chern}}(\vec{k}_{\text{2d}}, k_z) = \begin{pmatrix} V_{\alpha=0}(\vec{k}_{\text{2d}}, k_z) & 0 \\ 0 & V_{\alpha=1}(\vec{k}_{\text{2d}}, k_z) \end{pmatrix} \tag{33}$$

Focusing on each subspace for $\alpha = 0, 1$, each $2 \times 2$ matrix $V_{\alpha=0}(\vec{k}_{\text{2d}}, k_z), V_{\alpha=1}(\vec{k}_{\text{2d}}, k_z)$ is identified as pumping a p+ip superconductor [69]. To see this, one can check that the matrix $V_{\alpha=1}$ can be homotopically deformed into $V_{\alpha=0}$

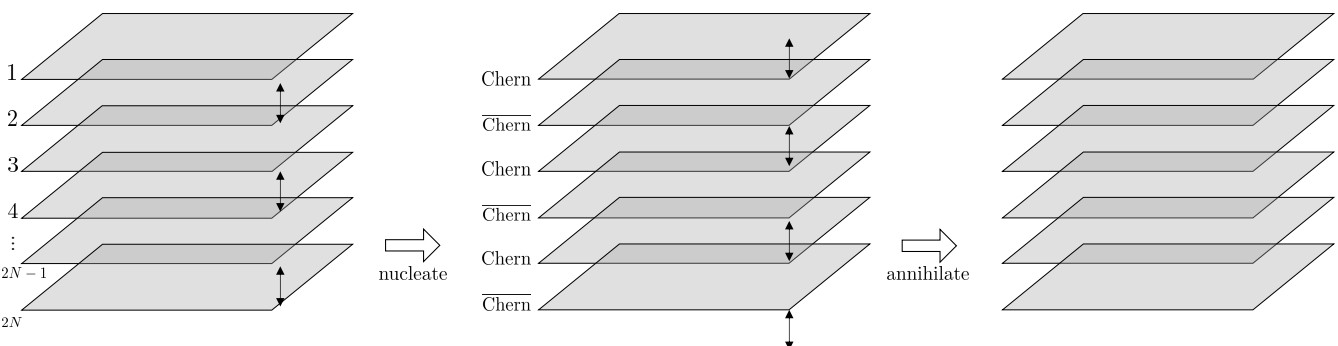

Figure 4: Pumping a Chern insulator through $2N$ layers of atomic insulators.

after performing the particle-hole conjugation acting on the $\alpha = 0$ energy band, since both of the unitaries $V_\alpha$ turn out to carry the same winding number 1 associated with the map $V_\alpha : T^3 \to SU(2) \cong S^3$. The unitary can then be regarded as two copies of the identical unitaries acting on each band $\alpha = 0, 1$, both together constitute the pumping of a Chern insulator. This means that each of $V_{\alpha=0}(\vec{k}_{2\mathrm{d}}, k_z), V_{\alpha=1}(\vec{k}_{2\mathrm{d}}, k_z)$ carries a "half" of the Chern insulator, which is a p+ip superconductor. Therefore, let us explicitly write the unitary pumping the p+ip superconductor as

$$V_{\mathrm{p+ip}} = \prod_{\vec{k}_{2\mathrm{d}}, k_z} V_{\alpha=0}(\vec{k}_{2\mathrm{d}}, k_z). \tag{34}$$

## 4.3  Fault-tolerant CCZ gate

Here we argue that the above unitaries pumping the Chern insulator or p+ip superconductor can be expressed as a constant-depth circuit of quasi-local unitaries with at most exponentially decaying tails, following [69]. The unitaries $U^{\mathrm{nucl}}_{2j-1,2j}(\vec{k}_{2\mathrm{d}}), U^{\mathrm{annih}}_{2j,2j+1}(\vec{k}_{2\mathrm{d}})$ can be expressed as the time-ordered exponential for the Hamiltonian evolution,

$$U^{\mathrm{nucl}}(\vec{k}_{2\mathrm{d}}) = \mathcal{T} \exp\left( i \int_0^t d\tau K^{\mathrm{nucl}}(\vec{k}_{2\mathrm{d}}, \tau) \right), \quad U^{\mathrm{annih}}(\vec{k}_{2\mathrm{d}}) = \mathcal{T} \exp\left( i \int_0^t d\tau K^{\mathrm{annih}}(\vec{k}_{2\mathrm{d}}, \tau) \right), \tag{35}$$

with $K^{\mathrm{nucl}}(\vec{k}_{2\mathrm{d}}, \tau) = -iU^{\mathrm{nucl}}(\vec{k}_{2\mathrm{d}}, \tau)^{-1}\partial_\tau U^{\mathrm{nucl}}(\vec{k}_{2\mathrm{d}}, \tau)$ a Hermitian operator. Here, a unitary $U^{\mathrm{nucl}}(\vec{k}_{2\mathrm{d}}, \tau)$ is a function of $0 \le \tau \le t$ that smoothly interpolates between $U^{\mathrm{nucl}}(\vec{k}_{2\mathrm{d}}, \tau = 0) = \mathrm{id}$ and $U^{\mathrm{nucl}}(\vec{k}_{2\mathrm{d}}, \tau = t) = U^{\mathrm{nucl}}(\vec{k}_{2\mathrm{d}})$. Such a smooth interpolation is possible, because $U^{\mathrm{nucl}}(\vec{k}_{2\mathrm{d}})$ is identified as a map $T^2 \to SU(4)$ which can be homotopically deformed to an identity map since $\pi_1(SU(4)) = \pi_2(SU(4)) = 0$. A similar description also works for $K^{\mathrm{annih}}(\vec{k}_{2\mathrm{d}}, \tau)$.

Since the Hamiltonians $K^{\mathrm{nucl}}(\vec{k}_{2\mathrm{d}}, \tau), K^{\mathrm{annih}}(\vec{k}_{2\mathrm{d}}, \tau)$ are smooth functions of $\vec{k}_{2\mathrm{d}}, \tau$, this evolution is quasi-local in real space in the sense that they have at most exponentially decaying tails. This implies that the pumping of the Chern insulator $U_{\mathrm{Chern}}$ can be implemented by a finite-time evolution by a quasi-local Hamiltonian with exponentially decaying tails. We expect that this can be approximated by a constant depth local unitary circuit and it can give a fault-tolerant logical operation, but this requires further study to understand in detail.

The unitary $U_{\mathrm{Chern}}$ acts within the Fock space of each flavor $\alpha = 0, 1$ with $Z^{\mathrm{fl}} = (-1)^\alpha$, and focusing on one flavor index we obtain the operator $V_{\alpha=0}, V_{\alpha=1}$ pumping the p+ip superconductor. By gauging the $\mathbb{Z}_2^f$ symmetry for each flavor distinctly, we obtain two copies of (3+1)D fermionic toric codes, where the local constant-depth circuit $U_{\mathrm{Chern}}$ acts as the diagonal fault-tolerant logical gate $V_{\mathrm{p+ip}} \otimes V_{\mathrm{p+ip}}$ on the copy.

The pumping of p+ip superconductor $V_{\mathrm{p+ip}}$ realizes the logical $CCZ$ gate on a torus $T^3$ [69]. This is understood as a consequence of the permutation of 1-form symmetry generators by the action of the p+ip symmetry defect discussed in Sec. 2.2, leading to the group commutation relation between the emergent symmetries introduced in Sec. 3,

$$\text{Permutation action:} \quad [V_{\mathrm{p+ip}}, X_{yz}] = \mathcal{W}_{\mathrm{K},yz}, \tag{36}$$

which corresponds to the permutation action where the magnetic surface operator $X_{yz}$ extended in $y, z$ direction is attached to the surface operator for the Kitaev chain. As discussed in Sec. 3.1, the Kitaev chain operator $\mathcal{W}_{\mathrm{K},yz}$ on a torus $T^2_{yz}$ implements the $CZ$ gate for the logical qubits on $T^3$ with Pauli operators $\{Z_y, X_{zx}\}, \{Z_z, X_{xy}\}$. The above commutation relation is realized by $V_{\mathrm{p+ip}} = CCZ$ involving all three logical qubits of the $\mathbb{Z}_2$ fermionic toric code on $T^3$.

# 5 Example: fault-tolerant controlled-S gate in (3+1)D fermionic $\mathbb{Z}_2$ toric code

Next let us consider the 3d space $M^3 = T^2 \rtimes_{\mathcal{C}} S^1$, which is a mapping torus of $T^2$ twisted by the modular transformation $\mathcal{S}^2 = \mathcal{C}$. As we will see below, pumping the p+ip superconductor through this mapping torus can implement a fault-tolerant Controlled-$S^\dagger$ gate.

## 5.1 Warm-up: Controlled-Z gate by pumping Chern insulator and 3-group

Before discussing the properties of the p+ip defect, let us first consider the $2\mathbb{Z}$ subgroup of the 0-form symmetry generated by the Chern insulator, generated by the unitary $V_{\text{p+ip}}^2$. As described in Sec. 2.4, this 0-form symmetry for the Chern insulator forms a non-trivial 3-group together with the other emergent invertible symmetries. The 3-group structure is characterized by the relation among the background gauge fields

$$dC_3 = Sq^2(C_2) + B_2 \cup C_2 + \left(\frac{d\tilde{B}_2}{2} + w_3\right) \cup C_1 + (B_2 + w_2) \cup \frac{d\tilde{C}_1}{4} \ , \tag{37}$$

where the backgrounds of 0-form symmetry, magnetic 1-form symmetry $C_1 \in Z^1(M_4, \mathbb{Z})$, 1-form symmetry for the Kitaev chain, and electric 2-form symmetry are denoted by $C_1 \in Z^1(M_4, \mathbb{Z})$, $B_2 \in Z^2(M_4, \mathbb{Z}_2)$, $C_2 \in Z^2(M_4, \mathbb{Z}_2)$, $C_3 \in C^3(M_4, \mathbb{Z}_2)$. The last term for the 3-group $\frac{d\tilde{B}_2}{2} \cup C_1$ affects on the commutation relation between 0-form symmetry generator $V_{\text{p+ip}}^2$ and the magnetic surface operator $X$. To see this effect explicitly, let us consider a 3d space given by a mapping torus $M^3 = T^2 \rtimes_{\mathcal{C}} S^1$. This 3d space can be filled by a cubic lattice with the boundary condition along $z$ direction twisted by $C_2$ rotation of the square lattice, which makes up $T_{xy}^2 \rtimes_{C_2} S_z^1$ (see Figure 5 (a)). This topology stores three logical qubits acted on by three pairs of the Pauli operators $\{X_{xy}, Z_z\}, \{X_{yz}, Z_x\}, \{X_{zx}, Z_y\}$.

In this setup, the magnetic surface $X_{yz}$ extended in $yz$ direction is supported on a Klein bottle. Due to the above non-trivial 3-group structure, the 0-form symmetry for the Chern insulator acts on the junction of the magnetic defects by attaching an electric charge. Noting that the junction here corresponds to the orientation-reversing defect of the magnetic surface, the 3-group structure is reflected in the commutation relation between the generator of 0-form and 1-form symmetry giving that of the 2-form symmetry,

$$\text{3-group equation:} \quad [V_{\text{p+ip}}^2, X_{yz}] = Z_y. \tag{38}$$

This implies that $V_{\text{p+ip}}^2$ acts by the $CZ$ gate on the two logical qubits with Pauli operators $\{X_x, Z_{yz}\}, \{X_y, Z_{xz}\}$ encoded in the $\mathbb{Z}_2$ gauge theory. This $CZ$ gate comes from the second term of the expression of $V_{\text{p+ip}}$ in Eq. (26) that involves Controlled-$S^\dagger$ between two qubits, where we get $CZ$ from the square of $CS^\dagger$.

## 5.2 Lattice model for pumping Chern insulator through $T^2 \rtimes_{C_2} S^1$

One can explicitly construct a unitary $V_{\text{p+ip}}^2$ on a cubic lattice defined on $T_{xy}^2 \rtimes_{C_2} S_z^1$. This can be done by slightly generalizing a unitary pumping a Chern insulator obtained in [69] to the case with twisted boundary condition by $C_2$ along the $z$ direction.

In the presence of the $C_2$ twisted boundary condition, we want the pumped Chern insulator to be symmetric under the $C_2$ rotation. Suppose that the $C_2$ symmetry acts on the complex fermion $\{f^\alpha, f^{\alpha\dagger}\}$ of the Chern insulator by

$$f^\alpha[\vec{k}_{\text{2d}}] \rightarrow (Z^{\text{fl}})_{\alpha\beta} \cdot f^\beta[-\vec{k}_{\text{2d}}], \tag{39}$$

one can then see that this $C_2$ action makes the Hamiltonian of the Chern insulator $H_{\text{Chern}}$ explicitly defined in Appendix C. In the presence of the $C_2$ twisted boundary condition, one can also define the unitary pumping the Chern insulator in the same fashion as the untwisted case,

$$U_{\text{Chern}}[T^2 \rtimes_{C_2} S^1] = \left(\prod_{1 \le j \le N} U_{2j, 2j+1}^{\text{annih}}\right) \left(\prod_{1 \le j \le N} U_{2j-1, 2j}^{\text{nucl}}\right). \tag{40}$$

To see directly how the $C_2$ twisted boundary condition enables us to realize the Controlled-Z gate by pumping the Chern insulator, let us evaluate the action of the pumping unitary $U_{\text{Chern}}$ on the 3d trivial atomic insulator. This

unitary acts by a phase on the atomic insulator given by

$$\langle \text{trivial}| \, U_{\text{Chern}}[T^2 \rtimes_{C_2} S^1] \, |\text{trivial}\rangle = \langle \text{trivial}| \left( \prod_{1 \leq j \leq N} U^{\text{annih}}_{2j, 2j+1} \right) \cdot \left( \prod_{1 \leq j \leq N} U^{\text{nucl}}_{2j-1, 2j} \right) |\text{trivial}\rangle$$

$$= \langle \text{Chern}[2N+1] | \text{Chern}[1]\rangle \cdot \prod_{2 \leq j \leq 2N} \langle \text{Chern}[j] | \text{Chern}[j]\rangle \tag{41}$$

where the state $|\text{Chern}[j]\rangle$ is the Chern insulator state created at the $j$-th layer. Noting that the $C_2$ twisted boundary condition is introduced by the identification between $(2N+1)$-th layer and the 1st layer by the $C_2$ rotation, we can further rewrite the expression as

$$\langle \text{trivial}| \, U_{\text{Chern}}[T^2 \rtimes_{C_2} S^1] \, |\text{trivial}\rangle = \langle \text{Chern}[1] | C_2 | \text{Chern}[1]\rangle \cdot \prod_{2 \leq j \leq 2N} \langle \text{Chern}[j] | \text{Chern}[j]\rangle$$

$$= \frac{\langle \text{Chern}[1]| \, C_2 \, |\text{Chern}[1]\rangle}{\langle \text{Chern}[1] | \text{Chern}[1]\rangle} \cdot \langle \text{trivial}| \, U_{\text{Chern}}[T^3] \, |\text{trivial}\rangle , \tag{42}$$

where the last equation shows that the difference between the unitaries $U_{\text{Chern}}[T^2 \rtimes_{C_2} S^1]$ and $U_{\text{Chern}}[T^3]$ is encoded in the layer on which the $C_2$ twist acts, where we have the $C_2$ expectation value at the single Chern insulator. We will see that this $C_2$ expectation value gives rise to a non-trivial logical gate.

For the free fermion Hamiltonian $H_{\text{Chern}}$ of the Chern insulator, one can explicitly evaluate the $C_2$ expectation value for each boundary condition of $\mathbb{Z}_2^f$ symmetry. For simplicity, let us assume that the Chern insulator is defined on a periodic square lattice with the length $L_x, L_y$ both even. In that case, the $C_2$ expectation value taken on the Chern insulator with each boundary condition is given by

$$\langle \text{AP}, \text{AP}| \, C_2 \, |\text{AP}, \text{AP}\rangle = \langle \text{P}, \text{AP}| \, C_2 \, |\text{P}, \text{AP}\rangle = \langle \text{AP}, \text{P}| \, C_2 \, |\text{AP}, \text{P}\rangle = 1,$$

$$\langle \text{P}, \text{P}| \, C_2 \, |\text{P}, \text{P}\rangle = \prod_{\vec{k}_{2d} = -\vec{k}_{2d}} \left\langle \vec{k}_{2d} \middle| C_2 \middle| \vec{k}_{2d} \right\rangle = -1. \tag{43}$$

where $\{\text{AP}, \text{P}\}$ denotes the $\mathbb{Z}_2^f$ boundary condition of the Chern insulator in $x$, $y$ direction. The above relation can be checked as follows. First, note that the $C_2$ acts on the pair of complex fermions $c_{\vec{k}}^\dagger c_{-\vec{k}}^\dagger$ with $\vec{k} \neq -\vec{k}$ by flipping the sign. When both of $L_x, L_y$ are even, there are always even number of such pairs, so one can discard the contribution of these pairs in the $C_2$ expectation value. Hence, only the high symmetric momentum under $C_2$ satisfying $\vec{k} = -\vec{k}$ contributes to the expression. Such high symmetric points exist only for $\{\text{P}, \text{P}\}$ boundary condition. By evaluating the $Z^{\text{fl}}$ eigenvalue at each high symmetric momentum of $H_{\text{Chern}}$, one can see that $\langle \text{P}, \text{P}| \, C_2 \, |\text{P}, \text{P}\rangle = -1$. This observation that this $C_2$ expectation value becomes $-1$ when the 2d torus $T^2$ has the periodic boundary condition in both $x$ and $y$ direction, and 1 otherwise, implies that $U_{\text{Chern}}[T^2 \rtimes_{C_2} S^1]$ in Eq. (42) implements the CZ gate with respect to the two logical qubits $\{X_x, Z_{yz}\}$, $\{X_y, Z_{xz}\}$.

At the level of effective field theory of the Chern insulator, one can also see that the $C_2$ expectation value gives the sign that corresponds to the CZ gate. To see this, we describe the state of the Chern insulator on $T^2$ in terms of the $U(1)_4$ Chern-Simons theory, which is the bosonic dual of the Chern insulator under the sixteen fold way. The $U(1)_4$ theory has four anyons $\{1, v, \psi, v\psi\}$ including the trivial one, so it has four-dimensional Hilbert space on $T^2$, where each state is labeled by the anyon. The Chern insulator state for each boundary condition of $T^2$ is then expressed by that of the $U(1)_4$ as

$$|\text{AP}, \text{AP}\rangle = |1\rangle + |\psi\rangle ,$$
$$|\text{AP}, \text{P}\rangle = |1\rangle - |\psi\rangle ,$$
$$|\text{P}, \text{AP}\rangle = |v\rangle + |v\psi\rangle , \tag{44}$$
$$|\text{P}, \text{P}\rangle = |v\rangle - |v\psi\rangle ,$$

The $C_2$ rotation now acts by the $\mathcal{S}^2$ modular transformation on the states of $U(1)_4$ theory, which turns to be the charge conjugation permuting the anyons as $v \leftrightarrow v\psi$. Indeed, $\mathcal{S}, \mathcal{S}^2$ matrices are expressed in the basis of $\{|1\rangle, |v\rangle, |\psi\rangle, |v\psi\rangle\}$ as

$$\mathcal{S} = \frac{1}{2} \begin{pmatrix} 1 & 1 & 1 & 1 \\ 1 & -i & -1 & i \\ 1 & -1 & 1 & -1 \\ 1 & i & -1 & -i \end{pmatrix}, \quad \mathcal{S}^2 = \begin{pmatrix} 1 & & & \\ & & & 1 \\ & & 1 & \\ & 1 & & \end{pmatrix} \tag{45}$$

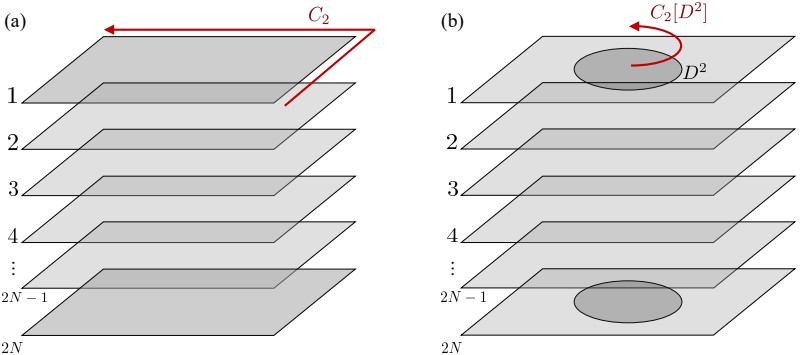

Figure 5: Boundary conditions of the layered system. (a): Twisted boundary condition for $C_2$ rotation symmetry acting on the whole 2d torus, which makes the topology of $T^2 \rtimes_{C_2} S^1$. (b): Twisted boundary condition for the partial $C_2$ rotation acting on the disk within the 2d torus, which makes the topology of $\mathbb{RP}^3 \# T^3$.

One can then immediately see that $\mathcal{S}^2$ acts on the state $|\mathrm{P}, \mathrm{P}\rangle$ by $-1$ sign, while acts as identity on the others. This gives the alternative field theoretical explanation for why the above unitary $U_{\mathrm{Chern}}[T^2 \rtimes_{C_2} S^1]$ implements the CZ gate. [8]

## 5.3 Controlled-S gate by pumping p+ip superconductor

Let us then study the logical gate implemented by pumping the p+ip superconductor. As we have seen in Sec. 2.2, the codimension-1 p+ip defect induces the permutation of the generators of the 1-form symmetry, by attaching the Kitaev chain defect to the magnetic defect. This permutation action is expressed as the commutation relation between the 0-form and 1-form symmetry generators on $T^2 \rtimes_{C_2} S^1$,

$$\text{Permutation action:} \quad [V_{\mathrm{p+ip}}, X_{yz}] = \mathcal{W}_{\mathrm{K},yz}. \tag{46}$$

The expression of the Kitaev chain operator $\mathcal{W}_{\mathrm{K},yz}$ as a logical gate can be read from Eq. (25), and given by

$$\mathcal{W}_{\mathrm{K},yz} = CZ_{y,z} CZ_{x,y} S_y^\dagger \tag{47}$$

we then have

$$V_{\mathrm{p+ip}} = CCZ_{x,y,z} CS_{x,y}^\dagger, \tag{48}$$

where the Controlled-$S^\dagger$ logical gate $CS_{x,y}^\dagger$ is enabled by the $C_2$ twisted boundary condition. To explicitly see how the $C_2$ twist makes the pumping of p+ip superconductor a non-trivial logical operation, we again consider the 3d system that consists of the $2N$ layers of the 2d trivial atomic insulators, and suppose of the process nucleating the pair of p+ip and p−ip superconductors for the pair of layers $(2j - 1, 2j)$, then annihilate them for each pair $(2j, 2j + 1)$. By repeating the same argument as the previous subsection, the unitary realizing the whole pumping process $V_{\mathrm{p+ip}}$ acts on the 3d trivial insulator by a phase, which is given by

$$\langle \text{trivial}| V_{\mathrm{p+ip}}[T^2 \rtimes_{C_2} S^1] |\text{trivial}\rangle = \frac{\langle \mathrm{p+ip}| C_2 |\mathrm{p+ip}\rangle_1}{\langle \mathrm{p+ip}|\mathrm{p+ip}\rangle_1} \cdot \langle \text{trivial}| V_{\mathrm{p+ip}}[T^3] |\text{trivial}\rangle, \tag{49}$$

where $\langle \mathrm{p+ip}| C_2 |\mathrm{p+ip}\rangle_1$ is the $C_2$ expectation value of the p+ip superconductor at the first layer.

We can again understand the origin of the Controlled-$S^\dagger$ gate at the level of the effective field theory of p+ip superconductor. To do this, we describe the state of the p+ip superconductor on $T^2$ by that of the Ising TQFT with the anyons $\{1, \sigma, \psi\}$, which is the bosonic dual of p+ip superconductor under the sixteen fold way. The state of p+ip superconductor with each boundary condition is then expressed as [93]

$$\begin{aligned}
|\mathrm{AP}, \mathrm{AP}\rangle &= |1\rangle + |\psi\rangle, \\
|\mathrm{AP}, \mathrm{P}\rangle &= |1\rangle - |\psi\rangle, \\
|\mathrm{P}, \mathrm{AP}\rangle &= |\sigma\rangle, \\
|\mathrm{P}, \mathrm{P}\rangle &= |\sigma; \psi\rangle,
\end{aligned} \tag{50}$$

---

[8]In general, the action of the crystalline $C_2$ rotation can differ from the modular $\mathcal{C}$ transformation of effective field theory by an internal $\mathbb{Z}_2$ symmetry, as a result of the crystalline equivalence principle [91, 92]. Our setup corresponds to the case that the internal $\mathbb{Z}_2$ symmetry acts trivially on the Hilbert space, where the microscopic $C_2$ action is identified as the modular transformation. It would be interesting to study how the different action of crystalline $C_2$ symmetry gives rise to a distinct realization of the logical gates.

where $|\sigma;\psi\rangle$ is the state on a punctured torus, where the fermion $\psi$ is inserted at the puncture and terminates at the $\sigma$ line in the bulk of the solid torus. The modular $\mathcal{S}$ matrix of the punctured torus is computed in [93], and given by $\mathcal{S}_{\mathrm{P,P}} = e^{7\pi i/4}$. The $\mathcal{S}^2$ acts on the other sectors in a trivial way, which is written in the basis of $\{|1\rangle, |\psi\rangle, |\sigma\rangle\}$ as

$$\mathcal{S} = \frac{1}{2}\begin{pmatrix} 1 & 1 & \sqrt{2} \\ 1 & 1 & -\sqrt{2} \\ \sqrt{2} & -\sqrt{2} & 0 \end{pmatrix}, \quad \mathcal{S}^2 = \mathrm{id}. \tag{51}$$

The $C_2$ rotation $\mathcal{S}^2$ hence acts by the phase $(-i)$ in the $\{\mathrm{P,P}\}$ sector, while it acts trivially on the other sector. This means the logical gate involving two logical qubits $\{X_x, Z_{yz}\}$, $\{X_y, Z_{xz}\}$ is given by $\mathrm{diag}(1,1,1,-i)$ in the $Z$ basis, which is nothing but the Controlled-$S^\dagger$ gate.

## 5.4 Fault-tolerance of Controlled-S gate

Following the argument reviewed in Sec. 4.3, one can immediately see that the pumping unitary $U_{\mathrm{Chern}}[T^2 \rtimes_{C_2} S^1]$ for the Chern insulator implements the fault-tolerant logical gate. Again, both of the unitaries $U^{\mathrm{nucl}}_{2j-1,2j}(\vec{k}_{2\mathrm{d}})$, $U^{\mathrm{annih}}_{2j,2j+1}(\vec{k}_{2\mathrm{d}})$ can be regarded as a finite-time evolution by the quasi-local Hamiltonian with at most exponentially decaying tail regardless of the $C_2$ twisted boundary condition. This unitary $U_{\mathrm{Chern}}[T^2 \rtimes_{C_2} S^1]$ again acts within the Fock space of each flavor $\alpha = 0, 1$ with $Z^{\mathrm{fl}} = (-1)^\alpha$, and focusing on one flavor index we obtain the operator $V_{\alpha=0}, V_{\alpha=1}$ pumping the p+ip superconductor, both of which implement the Controlled-$S^\dagger$ gate. By gauging the $\mathbb{Z}_2^f$ symmetry for each flavor distinctly, we obtain two copies of (3+1)D fermionic toric codes, where the local constant-depth circuit $U_{\mathrm{Chern}}[T^2 \rtimes_{C_2} S^1]$ then acts as the diagonal Controlled-$S^\dagger$ gate $V_{\mathrm{p+ip}} \otimes V_{\mathrm{p+ip}}$ on the copy.

# 6 Example: T gate in (3+1)D fermionic $\mathbb{Z}_2$ toric code

Finally we consider $M^3 = \mathbb{RP}^3$, where pumping p+ip superconductor through $M^3$ is expected to realize the logical $T$ gate according to Eq. (26). In order to realize the topology of $\mathbb{RP}^3$ in a lattice model, we consider a boundary condition of a 3d layered system twisted by "partial" $C_2$ rotation [94], as shown in Figure 5 (b). To understand how the partial rotation realizes the $\mathbb{RP}^3$ topology, we recall that $\mathbb{RP}^3$ is obtained by carving out a disk $D^3$ from a 3-sphere, and then identifying the north and south hemisphere of the boundary $S^2$ by the $C_2$ rotation map. The boundary condition implemented by the partial $C_2$ rotation exactly does the identification of hemispheres to obtain $\mathbb{RP}^3$. In the presence of this boundary condition, the 3d space for the layered system has the topology of $T^3 \# \mathbb{RP}^3$, where the partial rotation adds the $\mathbb{RP}^3$ to the original torus topology $T^3$.

In order to construct an explicit logical $T$ gate in this topology by a local constant-depth circuit, we would need to obtain a unitary circuit which can nucleate a "bubble" of a p+ip superconductor enclosing the partial rotation cross-cap. Then, one would be able to sweep the p+ip superconductor over the whole 3d space, resulting in an emergent symmetry acting on the code space of fermionic $\mathbb{Z}_2$ toric code. While we do not attempt to work out the construction of such a unitary logical gate pumping a (2+1)D invertible phase through the partial rotation cross-cap, we strongly expect that a constant-depth unitary circuit for this logical gate exists. Indeed, in (2+1)D one can pump a (1+1)D Kitaev chain through a cross-cap for spatial reflection by a local constant-depth circuit to obtain the logical gate of the (2+1)D $\mathbb{Z}_2$ toric code on $\mathbb{RP}^2$ [62]. A similar construction would work for pumping a (2+1)D invertible phase through a cross-cap of $\mathbb{RP}^3$ as well.

In this section, instead of explicitly constructing a logical $T$ gate using a local unitary circuit, we comment on our expectation for a certain operator which is unitary in the whole Hilbert space of the 3d layered system, but does not act within the ground state Hilbert space, so does not directly define the logical gate. Nevertheless, after projecting onto the code space of fermionic $\mathbb{Z}_2$ toric code, it is expected to become a non-unitary operator proportional to the logical $T$ gate up to overall amplitude.

## 6.1 Warm-up: S gate by pumping Chern insulator and 3-group

Before considering the p+ip case, let us discuss the logical gate implemented by the Chern insulator defect, which generates the $2\mathbb{Z}$ subgroup of the 0-form symmetry. Due to the 3-group structure $dC_3 = d\tilde{B}_2/2 \cup C_1$ discussed in Sec. 2.4, the 0-form symmetry for the Chern insulator acts on the junction of the magnetic defects by attaching the electric charge. As discussed in Sec. 5, this implies that the 0-form symmetry acts on the orientation-reversing defect of the magnetic defect by attaching the Wilson line of the electric charge.

Let us now consider a 3d space $\mathbb{RP}^3$, where we can think of the magnetic surface operator supported at $\mathbb{RP}^2$ embedded in $\mathbb{RP}^3$. This surface operator implements the logical $X$ gate of the $\mathbb{Z}_2$ gauge theory. The 3-group

structure discussed above is then reflected in the commutation relation between the 0-form and 1-form symmetry generator,

$$\text{3-group equation:} \quad [V_{\mathrm{p+ip}}^2, X] \propto Z, \tag{52}$$

where Pauli $Z$ corresponds to the Wilson line of a fermion extended along the orientation-reversing cycle of $\mathbb{RP}^2$, which is a nontrivial cycle of $\mathbb{RP}^3$. This commutation relation is indeed realized by Eq. (26), which says $V_{\mathrm{p+ip}}^2 = S$.

## 6.2 Realizing $\mathbb{RP}^3$ topology via partial rotation, and pumping Chern insulator through $\mathbb{RP}^3$

Here we consider the process of pumping the Chern insulator discussed in previous subsections on a layered lattice system defined on a 3d space with the topology of $\mathbb{RP}^3$. As we mentioned at the beginning of the section, to encode the topology of $\mathbb{RP}^3$ in the 3d space, we employ the idea of partial rotation, which is the spatial rotation acting within a disk subregion. For the system of $2N$ layers of the 2d trivial atomic insulators, we consider the boundary condition that identifies the $(2N+1)$-th layer with the first layer by the partial $C_2$ rotation acting on the disk inside the 2d torus (see Figure 5 (b)). The partial-rotation twist works as a "cross-cap" making up the $\mathbb{RP}^3$ topology, and the resulting 3d layered system corresponds to a discretization of the 3-manifold $T^3 \# \mathbb{RP}^3$.

Let us study the action of the operator pumping the Chern insulator in the presence of the partial $C_2$ rotation twist. We can just define the pumping unitary in the same form as Eq. (32),

$$U_{\mathrm{Chern}}[T^3 \# \mathbb{RP}^3] = \left( \prod_{1 \leq j \leq N} U_{2j,2j+1}^{\mathrm{annih}} \right) \left( \prod_{1 \leq j \leq N} U_{2j-1,2j}^{\mathrm{nucl}} \right). \tag{53}$$

In the presence of the partial $C_2$ twist, the state for the trivial atomic insulator is no longer an eigenstate of the pumping operator $U_{\mathrm{Chern}}$, so this operator does not directly give the logical gate of $\mathbb{Z}_2$ gauge theory after gauging $\mathbb{Z}_2^f$ symmetry. However, after gauging $\mathbb{Z}_2^f$ fermion parity symmetry, we can see that the projection of $U_{\mathrm{Chern}}$ onto the ground state subspace for the $\mathbb{Z}_2$ gauge theory behaves as the logical $S$ gate on the logical qubit encoded by the $\mathbb{RP}^3$ topology. To check this, let us explicitly compute the expectation value of the unitary $U_{\mathrm{Chern}}$ on the trivial atomic insulator,

$$\langle \mathrm{trivial} | U_{\mathrm{Chern}}[T^3 \# \mathbb{RP}^3] | \mathrm{trivial} \rangle = \frac{\langle \mathrm{Chern} | C_2[D^2] | \mathrm{Chern} \rangle_1}{\langle \mathrm{Chern} | \mathrm{Chern} \rangle_1} \cdot \langle \mathrm{trivial} | U_{\mathrm{Chern}}[T^3] | \mathrm{trivial} \rangle, \tag{54}$$

where $C_2[D^2]$ is the partial $C_2$ rotation operator acting within the disk of the first layer. One can also get the expectation value of the operator $U_{\mathrm{Chern}}$ in the presence of the $\mathbb{Z}_2^f$-twisted boundary condition for $\mathbb{RP}^3$, which corresponds to shifting the spin structure of $\mathbb{RP}^3$. This $\mathbb{Z}_2^f$-twisted boundary condition is achieved by inserting the $\mathbb{Z}_2^f$ symmetry defect at the disk where we act by the partial rotation. The expectation value of $U_{\mathrm{Chern}}$ with the $\mathbb{Z}_2^f$-twisted boundary condition is then given by

$$\langle \mathrm{trivial} | U_{\mathrm{Chern}}[T^3 \# \mathbb{RP}^3, \mathrm{twisted}] | \mathrm{trivial} \rangle = \frac{\langle \mathrm{Chern} | C_2(-1)^F[D^2] | \mathrm{Chern} \rangle_1}{\langle \mathrm{Chern} | \mathrm{Chern} \rangle_1} \cdot \langle \mathrm{trivial} | U_{\mathrm{Chern}}[T^3] | \mathrm{trivial} \rangle, \tag{55}$$

where $C_2(-1)^F[D^2]$ is the partial $C_2$ rotation associated with the partial $\mathbb{Z}_2^f$ fermion parity operator, acting within the disk of the first layer. The expectation value of the partial rotation operator in the Chern insulator has been computed in [94], and has the form of

$$\langle \mathrm{Chern} | C_2[D^2] | \mathrm{Chern} \rangle_1 = e^{-\frac{2\pi i}{8}} \times \chi, \quad \langle \mathrm{Chern} | C_2(-1)^F[D^2] | \mathrm{Chern} \rangle_1 = e^{\frac{2\pi i}{8}} \times \chi, \tag{56}$$

where $\chi$ is a real non-universal value. See Appendix E for the detailed discussions about the evaluation of the partial rotation. One can hence see that the operator $U$ projected onto the state for a 3d atomic insulator is proportional to $S = \mathrm{diag}(1, e^{\frac{2\pi i}{4}})$ up to the overall value $e^{-2\pi i/8} \times \chi$.

## 6.3 Pumping p+ip superconductor through $\mathbb{RP}^3$ and T gate

Let us then study the logical gate implemented by pumping the p+ip superconductor in the presence of the partial $C_2$ twist. According to Eq. (26), the p+ip symmetry acting on a 3d space $\mathbb{RP}^3$ generates the logical $T$ gate on the

$\mathbb{Z}_2$ gauge theory. To explicitly see how the partial $C_2$ twist makes the pumping of p+ip superconductor a non-trivial logical operation, we again consider the process of nucleating the pair of p+ip and p−ip superconductors for the pair of layers $(2j-1, 2j)$, then annihilating them for each pair $(2j, 2j+1)$, in the system of $2N$ layers. This pumping unitary $V_{\text{p+ip}}$ does not act by unitary on the ground state subspace for the $\mathbb{Z}_2$ gauge theory, but acts by the logical $T$ gate once projected onto the ground state subspace.

Indeed, by repeating the same argument as the previous subsection, one can evaluate the expectation value of the pumping unitary on the 3d trivial atomic insulator as

$$\langle \text{trivial}| \, V_{\text{p+ip}}[T^3 \# \mathbb{RP}^3] \, |\text{trivial}\rangle = \frac{\langle \text{p+ip}| \, C_2[D^2] \, |\text{p+ip}\rangle_1}{\langle \text{p+ip}|\text{p+ip}\rangle_1} \cdot \langle \text{trivial}| \, V_{\text{p+ip}}[T^3] \, |\text{trivial}\rangle, \tag{57}$$

where $C_2[D^2]$ is the partial $C_2$ rotation operator acting within the disk of the first layer. Also, the expectation value with the shifted spin structure on $\mathbb{RP}^3$ is given by

$$\langle \text{trivial}| \, V_{\text{p+ip}}[T^3 \# \mathbb{RP}^3, \text{twisted}] \, |\text{trivial}\rangle = \frac{\langle \text{p+ip}| \, C_2(-1)^F[D^2] \, |\text{p+ip}\rangle_1}{\langle \text{p+ip}|\text{p+ip}\rangle_1} \cdot \langle \text{trivial}| \, V_{\text{p+ip}}[T^3] \, |\text{trivial}\rangle. \tag{58}$$

The computation details of the expectation value of the partial rotation operator in the p+ip superconductor is found in Appendix E, which largely follows the argument in [94, 95]. The result has the form of

$$\langle \text{p+ip}| \, C_2[D^2] \, |\text{p+ip}\rangle_1 = e^{-\frac{2\pi i}{16}} \times \chi, \quad \langle \text{p+ip}| \, C_2(-1)^F[D^2] \, |\text{p+ip}\rangle_1 = e^{\frac{2\pi i}{16}} \times \chi, \tag{59}$$

where $\chi$ is a real non-universal value. This demonstrates that the unitary becomes proportional to the logical $T$ gate $T = \text{diag}(1, e^{\frac{2\pi i}{8}})$ up to the overall value $e^{-2\pi i/16} \times \chi$.

# 7 Discussion

In this paper we have developed an understanding of the emergent 3-group symmetry of (3+1)D $\mathbb{Z}_2$ gauge theory with a fermionic charge. The 3-group symmetry is mathematically encoded by the equations obeyed by flat background gauge field configurations for the 3-group symmetry. Physically the 3-group symmetry describes the result of space-time intersections and junctions involving topological defects of varying codimension.

The emergent 3-group symmetry in general can act non-trivially in the code subspace of a topological code, and gives rise to fault-tolerant logical operations. In this paper we provided a general formula for the logical gates obtained by sweeping the Kitaev chain defect and the p+ip superconductor defect. When we consider the code defined on the torus $T^3$, the mapping torus $T^2 \rtimes S^1$, and $\mathbb{RP}^3$, our results imply that one can realize CCZ, CS, and T gates.

Our work raises several questions for further study. First, the unitary operators that we can define are in terms of finite-time evolution of a quasi-local Hamiltonian. It would be of interest to write down explicit constant depth local unitary circuits that approximate these unitaries and prove rigorously that they give fault-tolerant logical gates. For the $T$ gate on $\mathbb{RP}^3$, our microscopic realization required applying a partial $C_2$ rotation, which took the system out of the code subspace and required projecting back to the code subspace. As we discussed, we expect that there is a unitary operator that leaves the code subspace invariant and implements the same transformation; it would be of interest to explicitly find this operator.

Our results made use of the application of global and partial rotations to the topological states that are decorated on the codimension-1 defect. As studied in [94, 96, 95, 97], the results of these operations might be modified by non-trivial topological invariants protected by the crystalline rotation symmetry of the system, and may depend on the Wyckoff position of the fixed point of the rotation. It would be interesting to investigate whether and how these crystalline symmetry protected invariants affect the logical gates that are obtained.

As mentioned in the introduction, a large class of fault-tolerant logical gates can also be obtained in topological codes using mapping class group operations, which can be viewed as sweeping geometric defects through the system. The algebraic relationships between logical gates obtained from mapping class group elements and those obtained from sweeping invertible topological defects presumably give rise to a larger emergent 3-group symmetry that mixes the one studied here with the diffeomorphism symmetry of the code.

Finally, it would be interesting to place our results in the wider context of schemes for realizing universal fault-tolerant quantum computation, and whether these results be utilized to improve the optimal space-time overhead for universal quantum computation.

# Acknowledgement

We thank Guanyu Zhu and Zhenghan Wang for discussions, and Yuxuan Zhang for sharing unpublished results on numerical calculations of global and partial rotations in Chern insulators. RK thanks Ryan Thorngren, Weicheng Ye and Matthew Yu for discussions on Discord "Anomalology". We thank Xie Chen and Zhenghan Wang for comments on a draft. MB is supported by NSF CAREER grant (DMR- 1753240), and the Laboratory for Physical Sciences through the Condensed Matter Theory Center. PSH is supported by Simons Collaboration of Global Categorical Symmetries. RK is supported by JQI postdoctoral fellowship at the University of Maryland, and by National Science Foundation QLCI grant OMA-2120757.

# A    Derivations for logical gate of Kitaev chain defect

Here we derive the expression Eq. (25) of the logical gate implemented by the Kitaev chain defect. We will present two derivations, one based on property of the ABK invariant, the other based on commutation relation.

## A.1    Derivation from factorization property of invariant

The ABK invariant on the surface $\Sigma_j$ can be described by $\mathbb{Z}_2$ gauge field $b$ with the effective action [82, 44, 83, 36]

$$\frac{\pi}{2} \int_{\Sigma_j} q_a(b) \, , \tag{60}$$

where $b$ is a dynamical field that satisfies $db = 0$. Let us examine how the theory depends on the pin$^-$ structure $a$ by writing it as $a = a_0 + B$ for some fixed reference pin$^-$ structure $a_0$ (the pin$^-$ structure on the surface $\Sigma_j$ are the restriction of the spin structure in the 3-manifold $M$ that contains $\Sigma_j$ [44]). Using the property $q_{a_0+B}(x) = q_{a_0}(x) + 2x \cup B$ and $q(x + y) = q(x) + q(y) + 2x \cup y$, we can rewrite the effective action as

$$\frac{\pi}{2} \int_{\Sigma_j} q_{a_0}(b') - \frac{\pi}{2} \int_{\Sigma_j} q_{a_0}(B) \, , \tag{61}$$

where $b' = b + B$. Thus the theory depends on the pin$^-$ structure by the effective action

$$-\frac{\pi}{2} \int_{\Sigma_j} q_{a_0}(B) \, . \tag{62}$$

Consider the operator acting on the state $|\{a\}\rangle$ (where the eigenvalue of $(-1)^{\oint a} = \pm 1$ labels the spin structure of the 3-manifold $M$, which extends to pin$^-$ structure on the surface $\Sigma_j$ inside the 3-manifold) as follows:

$$\mathcal{W}_{\mathrm{K}}(\Sigma_j)|\{a\}\rangle = e^{-\frac{2\pi i}{4} \int_{\Sigma_j} q_{a_0}(a)}|\{a\}\rangle \, , \tag{63}$$

where $a_0$ is some fixed reference pin$^-$ structure related to the choice of embedding of the surface $\Sigma_j$ in the 3-manifold. For instance, if the spin structure differs from the pin$^-$ structure by $w_1$ of the surface, it is the same as the embedding with opposite orientation, since $\frac{\pi}{2} \int q_{a_0+w_1}(x) = \frac{\pi}{2} q_{a_0}(x) + \pi \int x \cup w_1 = -\frac{\pi}{2} \int q_{a_0}(x)$. It is sufficient to pick a choice of embedding; different choices are related by shifting $a$, $i.e.$ a basis transformation that conjugates $\mathcal{W}_K$ by Pauli $X$ gates. We can set $a_0 = 0$, but in the discussion let us keep general $a_0$.

Let us augment $\Sigma_j$ with additional surfaces such that $\{\Sigma_j, \Sigma_k\}$ forms a basis of $H_2(M, \mathbb{Z}_2)$. We can parameterize $a$ as

$$a = n_j \mathrm{PD}(\Sigma_j) + \sum_{k \neq j} n_k \mathrm{PD}(\Sigma_k) \, , \tag{64}$$

where $(-1)^{n_j}, (-1)^{n_k} = \pm 1$ are the eigenvalues of the Pauli $Z$ gate on the corresponding qubits (we will take them to be $n_j, n_k \in \{0, 1\}$), and PD denotes the Poincaré dual. In the notation of (25), $\mathrm{PD}(\Sigma_k) = \sigma_k$.

Let us substitute (64) into (63) and using the property $q(x + y) = q(x) + q(y) + 2x \cup y \mod 4$:

$$\mathcal{W}_{\mathrm{K}}(\Sigma_j)|\{a\}\rangle = \prod_{k<l, k, l \neq j} \mathrm{CZ}_{k,l}^{\int_{\Sigma_j} \sigma_k \cup \sigma_l} \left( \prod_{k \neq j} \mathrm{CZ}_{j,k}^{\int_{\Sigma_j} \sigma_j \cup \sigma_k} S_k^{-\int_{\Sigma_j} \sigma_k \cup \sigma_k} \right) S_j^{-\int_{\Sigma_j} \sigma_j \cup \sigma_j} |\{a\}\rangle \, , \tag{65}$$

where the CZ gates are from the contributions $\pi n_k n_l \int_{\Sigma_j} \sigma_k \cup \sigma_l$ and $\pi n_j n_k \int_{\Sigma_j} \sigma_j \cup \sigma_k$, the $S$ gates are from the contributions $n_k \frac{\pi}{2} \int_{\Sigma_j} q(\sigma_k)$ and $n_j \frac{\pi}{2} \int_{\Sigma_j} q(\sigma_j)$ where we used $n_j^2 = n_j, n_k^2 = n_k$ for $n_j, n_k = 0, 1$. Using the property

$$\int_{\Sigma_j} \sigma_j \cup \sigma_k = \int_M \sigma_j \cup \sigma_j \cup \sigma_k = \int_M Sq^1(\sigma_j) \cup \sigma_k = \int_M \sigma_j \cup Sq^1(\sigma_k) = \int_{\Sigma_j} \sigma_k \cup \sigma_k \mod 2 \ , \tag{66}$$

we can replace the term $\mathrm{CZ}_{j,k}^{\int_{\Sigma_j} \sigma_j \cup \sigma_k}$ in (65) with $\mathrm{CZ}_{j,k}^{\int_{\Sigma_j} \sigma_k \cup \sigma_k}$. Then the operator reproduces the expression (25) up to an overall constant phase $e^{\frac{2\pi i}{8} \#(\Sigma_j, \Sigma_j, \Sigma_j)}$.

## A.2    Derivation from commutation relation

Here we present an alternative derivation for Eq. (25) based on the commutation relations between the Kitaev chain operator and the logical Pauli operators. First, the commutation relation between the Kitaev chain operator and the logical Pauli $X$ operator is given by

$$\begin{aligned}
X_k \mathcal{W}_{\mathrm{K}}^\dagger(\Sigma_j) X_k \mathcal{W}_{\mathrm{K}}(\Sigma_j) \left|\{a\}\right\rangle &= \exp\left(\frac{2\pi i}{8}\left(\mathrm{ABK}(\Sigma_j, a) - \mathrm{ABK}(\Sigma_j, a + \sigma_k)\right)\right) \left|\{a\}\right\rangle \\
&= \exp\left(\frac{2\pi i}{4} \int_{\Sigma_j} q_a(\sigma_k)\right) \left|\{a\}\right\rangle .
\end{aligned} \tag{67}$$

The phase $\exp\left(\frac{2\pi i}{4} \int_{\Sigma_j} q_a(\sigma_k)\right)$ measures the pin$^-$ structure of a surface $\Sigma_k$ along a close loop $\gamma$ Poincaré dual to $\sigma_j \cup \sigma_k$. This implies that this phase is proportional to the action of the Wilson line of the fermion $\psi$ along $\gamma$, which can be expressed as a product of logical Pauli $Z$ operators as $\prod_l Z_l^{\int_{M^3} \sigma_j \sigma_k \sigma_l}$. Concretely, we have

$$X_k \mathcal{W}_{\mathrm{K}}^\dagger(\Sigma_j) X_k \mathcal{W}_{\mathrm{K}}(\Sigma_j) = i^{\int_{M^3} \sigma_j \sigma_k \sigma_k} \prod_l Z_l^{\int_{M^3} \sigma_j \sigma_k \sigma_l} . \tag{68}$$

The additional phase factor $i^{\int_{M^3} \sigma_j \sigma_k \sigma_k}$ reflects the property that the phase $\exp\left(\frac{2\pi i}{4} \int_{\Sigma_j} q_a(\sigma_k)\right)$ becomes $\pm i$ when $\gamma$ crosses through the orientation-reversing defect of the surface $\Sigma_j$ odd times, which happens when $\int_{M^3} \sigma_j \sigma_k \sigma_k = 1$. Meanwhile, the commutators between the Kitaev chain operator and the Pauli $Z$ operators become trivial,

$$Z_k \mathcal{W}_{\mathrm{K}}^\dagger(\Sigma_j) Z_k \mathcal{W}_{\mathrm{K}}(\Sigma_j) = 1. \tag{69}$$

The unitary is completely specified by the commutator between Pauli $X$ and $Z$ operators up to overall phase. In our case, the above commutators fix the form of the Kitaev chain operator up to phase as

$$\mathcal{W}_{\mathrm{K}}(\Sigma_j) = \prod_{\substack{k,l \\ k<l, j\neq k, j\neq l}} \mathrm{CZ}_{k,l}^{\int_{M^3} \sigma_j \sigma_k \sigma_l} \cdot \prod_{\substack{k \\ j\neq k}} \mathrm{CZ}_{j,k}^{\int_{M^3} \sigma_j \sigma_k \sigma_k} (S_k^\dagger)^{\int_{M^3} \sigma_j \sigma_k \sigma_k} \cdot (S_j^\dagger)^{\int_{M^3} \sigma_j \sigma_j \sigma_j} . \tag{70}$$

One can check this expression as follows.

1. When $j \neq k$, the commutation relation can be evaluated by

$$\left[X_k, \prod_{\substack{l,m \\ l<m, j\neq l, j\neq m}} \mathrm{CZ}_{l,m}^{\int_{M^3} \sigma_j \sigma_l \sigma_m}\right] = \prod_{\substack{l \\ l\neq j, l\neq k}} Z_l^{\int_{M^3} \sigma_j \sigma_k \sigma_l} \tag{71}$$

and

$$\left[X_k, \prod_{\substack{l \\ j\neq l}} \mathrm{CZ}_{j,l}^{\int_{M^3} \sigma_j \sigma_l \sigma_l} (S_l^\dagger)^{\int_{M^3} \sigma_j \sigma_l \sigma_l}\right] = Z_j^{\int_{M^3} \sigma_j \sigma_k \sigma_k} \times i^{\int_{M^3} \sigma_j \sigma_k \sigma_k} Z_k^{\int_{M^3} \sigma_j \sigma_k \sigma_k} . \tag{72}$$

where $[A, B] := A^{-1}B^{-1}AB$. These two commutation relations together produce Eq. (68).

2. When $j = k$, the nontrivial commutation relation comes from

$$\left[X_j, \prod_{\substack{l \\ j \neq l}} CZ_{j,l}^{\int_{M^3} \sigma_j \sigma_l \sigma_l}\right] = \prod_{\substack{l \\ j \neq l}} Z_l^{\int_{M^3} \sigma_j \sigma_l \sigma_l} \tag{73}$$

and

$$[X_j, (S_j^\dagger)^{\int_{M^3} \sigma_j \sigma_j \sigma_j}] = i^{\int_{M^3} \sigma_j \sigma_j \sigma_j} Z_j^{\int_{M^3} \sigma_j \sigma_j \sigma_j} \tag{74}$$

These two commutation relations together produce Eq. (68).

The additional phase factor appears when the surface $\Sigma_j$ satisfies $\int_{\Sigma_j} w_1^2 = 1$, in which case the ABK invariant $\exp\left(\frac{2\pi i}{8} \text{ABK}(\Sigma_j, a)\right)$ becomes the 8th root of unity. This happens when $\int_{M^3} \sigma_j^3 = 1$, where we must introduce a phase factor $\exp\left(\frac{2\pi i}{8} \int_{M^3} \sigma_j^3\right)$. Otherwise the eigenvalue of the above operator should match the ABK invariant of the surface $\Sigma_j$, since the ABK invariant of $\Sigma_j$ with the choice of spin structure such that $Z_k = 1$ for all $k$ is expected to be 1. So we get the expression (25),

$$\mathcal{W}_{\text{K}}(\Sigma_j) = \prod_{\substack{k,l \\ k < l, j \neq k, j \neq l}} CZ_{k,l}^{\int_{M^3} \sigma_j \sigma_k \sigma_l} \cdot \prod_{\substack{k \\ j \neq k}} CZ_{j,k}^{\int_{M^3} \sigma_j \sigma_k \sigma_k} (S_k^\dagger)^{\int_{M^3} \sigma_j \sigma_k \sigma_k} \cdot (e^{\frac{2\pi i}{8}} S_j^\dagger)^{\int_{M^3} \sigma_j \sigma_j \sigma_j}. \tag{75}$$

For example, when the 3d space $M^3$ is taken to be $\mathbb{RP}^3$, the fermionic toric code stores a single logical qubit. The Kitaev chain operator on $\mathbb{RP}^2$ embedded in $\mathbb{RP}^3$ can then implement $S^\dagger$ logical gate up to overall phase $e^{2\pi i/8}$, which originates from the last term in the above expression. Such a logical $S^\dagger$ gate in $(2+1)\text{D } \mathbb{Z}_2$ toric code and honeycomb Floquet code on $\mathbb{RP}^2$ has been explicitly constructed in [62]. [9]

# B  Derivations for logical gate of p+ip defect

Let us write the Pauli operators of the $\mathbb{Z}_2 \times \mathbb{Z}_2$ gauge theory as $\{Z_{\text{f},j}, X_{\text{f},j}\}, \{Z_{\text{f}',j}, X_{\text{f}',j}\}$, where $Z_{\text{f}}, Z_{\text{f}'}$ denote the Wilson line for fermions $\psi, \psi'$ respectively. Each $\mathbb{Z}_2$ gauge theory has the Kitaev chain defect, with corresponding operators denoted by $\mathcal{W}_{\text{K}}(\Sigma_j), \mathcal{W}_{\text{K}'}(\Sigma_j)$.

To describe the expression of the symmetry operator corresponding to the (p+ip)×(p−ip) defect, let us first express the $\mathbb{Z}_2 \times \mathbb{Z}_2$ gauge theory in a different basis, by redefining the Pauli operators as

$$\tilde{Z}_{\text{f},j} = Z_{\text{f},j}, \quad \tilde{Z}_{\text{b},j} = Z_{\text{f},j} Z_{\text{f}',j}, \quad \tilde{X}_{\text{f},j} = X_{\text{f},j} X_{\text{f}',j}, \quad \tilde{X}_{\text{b},j} = X_{\text{f}',j}. \tag{76}$$

The Pauli $\tilde{Z}_{\text{b},j}$ operator corresponds to the Wilson line for the bosonic particle $b := \psi\psi'$, so the above expression amounts to describing the $\mathbb{Z}_2 \times \mathbb{Z}_2$ gauge theory as stacking of $\mathbb{Z}_2$ gauge theory with a bosonic particle and that with a fermionic particle. In this basis, we can describe the eigenstate of the Pauli $Z$ operators as $|\{a, b\}\rangle$ with $a, b$ $\mathbb{Z}_2$ gauge fields on $M^3$, where the Wilson line of $a$ corresponds to $Z_{\text{f}}$, and that of $b$ corresponds to $Z_{\text{b}}$. On the eigenstate $|\{a, b\}\rangle$, the action of the operator $V$ that corresponds to the (p+ip)×(p−ip) defect is given by [98, 99]

$$V |\{a, b\}\rangle = \exp\left(\frac{2\pi i}{8} \beta_a(b)\right) |\{a, b\}\rangle := \exp\left(\frac{2\pi i}{8} \text{ABK}(\text{PD}(b), a)\right) |\{a, b\}\rangle \tag{77}$$

where $\text{PD}(b)$ denotes a 2d surface Poincaré dual to the $\mathbb{Z}_2$ gauge field $b$.

This operator $V$ can be expressed in terms of a certain non-Clifford unitary. To see this, we compute the commutation relation between $V$ and Pauli operators. First, we obviously have $[V, \tilde{Z}_{\text{f},j}] = [V, \tilde{Z}_{\text{b},j}] = 1$ since $V$ is diagonal in the $Z$ basis. Then, the commutator involving the Pauli $X$ operator can be computed by exploiting the following property of the invariant $\beta_a(b)$ [98],

$$\exp\left(\frac{2\pi i}{8} (\beta_a(b) + \beta_a(b') - \beta_a(b + b'))\right) = \exp\left(\frac{2\pi i}{4} \int_{\text{PD}(b)} q_a(b')\right) \tag{78}$$

---

[9] In [62], this logical gate for the Kitaev chain operator on $\mathbb{RP}^2$ is presented as $\sqrt{Y}^\dagger$ gate instead of $S^\dagger = \sqrt{Z}^\dagger$, since [62] works in the basis where the line operator of the fermion on $\mathbb{RP}^2$ is identified as a $Y$ gate instead of $Z$ gate (up to overall phase).

$$\exp\left(\frac{2\pi i}{8}(\beta_{a+\sigma}(b) - \beta_a(b))\right) = \exp\left(-\frac{2\pi i}{4}\int_{\mathrm{PD}(\sigma)} q_a(b)\right) \tag{79}$$

Here we introduced the function $q_a(b)$ defined on a pin$^-$ surface $\Sigma$. $a$ is pin$^-$ structure induced on the defect $\Sigma$, and $b$ is the flat $\mathbb{Z}_2$ gauge field for the bosonic electric particle. $q_a(b)$ is a $\mathbb{Z}_4$-valued quadratic function with the property

$$\int_\Sigma q_a(b) + \int_\Sigma q_a(b') = \int_\Sigma q_a(b+b') + 2\int_\Sigma b \cup b' \qquad \mathrm{mod}\ 4. \tag{80}$$

The commutator between $V$ and Pauli $X$ operators are then computed by

$$\tilde{X}_{\mathrm{f},k} V^\dagger \tilde{X}_{\mathrm{f},k} V \,|\{a,b\}\rangle = \exp\left(-\frac{2\pi i}{8}\beta_{a+\sigma_k}(b)\right) \exp\left(\frac{2\pi i}{8}\beta_a(b)\right) |\{a,b\}\rangle = \exp\left(\frac{2\pi i}{4}\int_{\Sigma_k} q_a(b)\right) |\{a,b\}\rangle \tag{81}$$

$$\begin{aligned}
\tilde{X}_{\mathrm{b},k} V^\dagger \tilde{X}_{\mathrm{b},k} V \,|\{a,b\}\rangle &= \exp\left(-\frac{2\pi i}{8}\beta_a(b+\sigma_k)\right) \exp\left(\frac{2\pi i}{8}\beta_a(b)\right) |\{a,b\}\rangle \\
&= \exp\left(\frac{2\pi i}{4}\int_{\Sigma_k} q_a(b)\right) \cdot \exp\left(-\frac{2\pi i}{8}\mathrm{ABK}(\Sigma_k, a)\right) |\{a,b\}\rangle
\end{aligned} \tag{82}$$

We then use the following relation between $q_a$ and the ABK invariant

$$\mathrm{ABK}(\Sigma, a) - \mathrm{ABK}(\Sigma, a+b) = 2\int_\Sigma q_a(b) \qquad \mathrm{mod}\ 8. \tag{83}$$

We then have

$$\begin{aligned}
\tilde{X}_{\mathrm{f},k} V^\dagger \tilde{X}_{\mathrm{f},k} V \,|\{a,b\}\rangle &= \exp\left(\frac{2\pi i}{8}\mathrm{ABK}(\Sigma_k, a)\right) \exp\left(-\frac{2\pi i}{8}\mathrm{ABK}(\Sigma_k, a+b)\right) |\{a,b\}\rangle \\
&= \mathcal{W}_{\mathrm{K}}(\Sigma_k)\mathcal{W}_{\mathrm{K}'}^\dagger(\Sigma_k) |\{a,b\}\rangle
\end{aligned} \tag{84}$$

$$\begin{aligned}
\tilde{X}_{\mathrm{b},k} V^\dagger \tilde{X}_{\mathrm{b},k} V \,|\{a,b\}\rangle &= \exp\left(-\frac{2\pi i}{8}\mathrm{ABK}(\Sigma_k, a+b)\right) |\{a,b\}\rangle \\
&= \mathcal{W}_{\mathrm{K}'}^\dagger(\Sigma_k) |\{a,b\}\rangle
\end{aligned} \tag{85}$$

One can then obtain the commutation relation between $V$ and Pauli operators in the original basis $X_{\mathrm{f}}, X_{\mathrm{f}'}$ as

$$X_{\mathrm{f},k} V^\dagger X_{\mathrm{f},k} V = \mathcal{W}_{\mathrm{K}}(\Sigma_k), \quad X_{\mathrm{f}',k} V^\dagger X_{\mathrm{f}',k} V = \mathcal{W}_{\mathrm{K}'}^\dagger(\Sigma_k). \tag{86}$$

By expressing $V$ in the form of $V_{\mathrm{f}} \otimes V_{\mathrm{f}'}$, the problem now reduces to finding a unitary $V_{\mathrm{f}}$ acting within the $\mathbb{Z}_2$ gauge theory with a single fermion, satisfying

$$Z_{\mathrm{f},k} V_{\mathrm{f}}^\dagger Z_{\mathrm{f},k} V_{\mathrm{f}} = 1, \quad X_{\mathrm{f},k} V_{\mathrm{f}}^\dagger X_{\mathrm{f},k} V_{\mathrm{f}} = \mathcal{W}_{\mathrm{K}}(\Sigma_k), \tag{87}$$

where $V_{\mathrm{f}}$ is regarded as the action of the p+ip defect. This commutation relation $X_{\mathrm{f},k} V_{\mathrm{f}}^\dagger X_{\mathrm{f},k} V_{\mathrm{f}} = \mathcal{W}_{\mathrm{K}}(\Sigma_k)$ explicitly shows that the p+ip operator $V_{\mathrm{f}}$ induces the automorphism of 1-form symmetries $X, \mathcal{W}_{\mathrm{K}}$, as discussed in Sec. 2.2.

Since the unitary is completely specified by the commutator between Pauli $X$ and $Z$ operators up to overall phase, it suffices to find a single unitary that satisfies the above commutation relations. We find an expression of such a unitary $V_{\mathrm{f}}$ (up to overall phase) as

$$V_{\mathrm{f}}(M^3) = \prod_{\substack{j,k,l \\ j<k<l}} (CCZ_{j,k,l})^{\int_{M^3} \sigma_j \sigma_k \sigma_l} \cdot \prod_{\substack{j,k \\ j<k}} (CS_{j,k})^{\int_{M^3} \sigma_j \sigma_k \sigma_k} \cdot \prod_j (T_j^\dagger)^{\int_{M^3} \sigma_j \sigma_j \sigma_j}. \tag{88}$$

One can check this expression by the following commutation relations:

$$\left[X_k, \prod_{\substack{l,m,n \\ l<m<n}} (CCZ_{l,m,n})^{\int_{M^3} \sigma_l \sigma_m \sigma_n}\right] = \prod_{\substack{l,m \\ k\neq l, k\neq m, l<m}} (CZ_{l,m})^{\int_{M^3} \sigma_k \sigma_l \sigma_m}, \tag{89}$$

$$\left[X_k, \prod_{\substack{l,m \\ l<m}} (CS_{l,m})^{\int_{M^3} \sigma_l \sigma_m \sigma_m}\right] = \prod_{\substack{l \\ k \neq l}} (CZ_{k,l})^{\int_{M^3} \sigma_k \sigma_l \sigma_l} (S_l^\dagger)^{\int_{M^3} \sigma_k \sigma_l \sigma_l}, \tag{90}$$

$$\left[X_k, \prod_l (T_l^\dagger)^{\int_{M^3} \sigma_l \sigma_l \sigma_l}\right] = (e^{\frac{2\pi i}{8}} S_k^\dagger)^{\int_{M^3} \sigma_k \sigma_k \sigma_k}. \tag{91}$$

These three commutation relations together produce $X_{\mathrm{f},k} V_\mathrm{f}^\dagger X_{\mathrm{f},k} V_\mathrm{f} = \mathcal{W}_\mathrm{K}(\Sigma_k)$.

# C  Details of unitary pumping a Chern insulator

Here we describe the details for the lattice model of a Chern insulator and a unitary that pumps it, following the definitions of [69]. As outlined in the main text, the Hamiltonian of the Chern insulator has the form of

$$H_{\mathrm{Chern}}(\vec{k}_{2\mathrm{d}}) = c^X(\vec{k}_{2\mathrm{d}}) X^\mathrm{fl} + c^Y(\vec{k}_{2\mathrm{d}}) Y^\mathrm{fl} + c^Z(\vec{k}_{2\mathrm{d}}) Z^\mathrm{fl} \tag{92}$$

To describe an explicit example of $\{c^X, c^Y, c^Z\}$ for a Chern insulator, we take two numbers $0 < k_1 < k_2 \ll 1$ and define a function $\theta(|\vec{k}_{2\mathrm{d}}|)$, which satisfies $\theta(|\vec{k}_{2\mathrm{d}}|) = 0$ for $|\vec{k}_{2\mathrm{d}}| \geq k_2$, $\theta(|\vec{k}_{2\mathrm{d}}|) = \pi$ for $|\vec{k}_{2\mathrm{d}}| \leq k_1$, and smoothly interpolates between them for $k_1 \leq |\vec{k}_{2\mathrm{d}}| \leq k_2$. Then $\{c^X, c^Y, c^Z\}$ is taken to be

$$(c^X, c^Y, c^Z) = \begin{cases} (0,0,-1) & \text{for } |\vec{k}_{2\mathrm{d}}| < k_1 \\ \left(\frac{k_x}{|\vec{k}_{2\mathrm{d}}|} \sin\left(\theta(|\vec{k}_{2\mathrm{d}}|)\right), \frac{k_y}{|\vec{k}_{2\mathrm{d}}|} \sin\left(\theta(|\vec{k}_{2\mathrm{d}}|)\right), \cos\left(\theta(|\vec{k}_{2\mathrm{d}}|)\right)\right) & \text{for } k_1 \leq |\vec{k}_{2\mathrm{d}}| \leq k_2, \\ (0,0,1) & \text{for } |\vec{k}_{2\mathrm{d}}| > k_2 \end{cases} \tag{93}$$

which realizes a non-trivial winding number 1 associated with a map $\{c^X, c^Y, c^Z\} : T^2 \to S^2$, hence defines a Chern insulator.

Next, we describe the details of the unitary that pumps a Chern insulator. Let us define $\phi(\vec{k}_{2\mathrm{d}}) = \tan^{-1}(k_y/k_x)$, and the unitaries for $|\vec{k}_{2\mathrm{d}}| \geq k_1$,

$$V_+(\vec{k}_{2\mathrm{d}}) = \exp\left(i\phi(\vec{k}_{2\mathrm{d}})(1 + Z^\mathrm{fl})/2\right) \exp\left(i\theta(|\vec{k}_{2\mathrm{d}}|) Y^\mathrm{fl}/2\right) \tag{94}$$

$$V_-(\vec{k}_{2\mathrm{d}}) = \exp\left(-i\phi(\vec{k}_{2\mathrm{d}})(1 + Z^\mathrm{fl})/2\right) \exp\left(i\theta(|\vec{k}_{2\mathrm{d}}|) Y^\mathrm{fl}/2\right) \tag{95}$$

Then, a unitary $U^\mathrm{nucl}$ that can nucleate a pair of Chern insulator and its opposite out of a trivial atomic insulator is given by

$$U^\mathrm{nucl}(\vec{k}_{2\mathrm{d}}) = \begin{cases} \left(|\vec{k}_{2\mathrm{d}}|/k_1 \exp\left(i\phi(\vec{k}_{2\mathrm{d}}) Z^\mathrm{layer}(1 + Z^\mathrm{fl})/2\right) + i\sqrt{1 - |\vec{k}_{2\mathrm{d}}|^2/k_1^2} X^\mathrm{layer}\right) iY^\mathrm{fl} & \text{for } |\vec{k}_{2\mathrm{d}}| < k_1 \\ V_+(\vec{k}_{2\mathrm{d}})(1 + Z^\mathrm{layer})/2 + V_-(\vec{k}_{2\mathrm{d}})(1 - Z^\mathrm{layer})/2 & \text{for } k_1 \leq |\vec{k}_{2\mathrm{d}}| \leq k_2 \\ r(|\vec{k}_{2\mathrm{d}}|) \exp\left(i\phi(\vec{k}_{2\mathrm{d}}) Z^\mathrm{layer}(1 + Z^\mathrm{fl})/2\right) + i\sqrt{1 - r(|\vec{k}_{2\mathrm{d}}|)^2/k_1^2} X^\mathrm{layer} & \text{for } |\vec{k}_{2\mathrm{d}}| > k_2, \end{cases} \tag{96}$$

where $r(|\vec{k}_{2\mathrm{d}}|)$ is a positive function satisfying $r(|\vec{k}_{2\mathrm{d}}|) = 1$ for $|\vec{k}_{2\mathrm{d}}| \leq k_2$, $r(|\vec{k}_{2\mathrm{d}}|) = 0$ for $|\vec{k}_{2\mathrm{d}}| \geq 1$, and smoothly interpolates between them for $k_2 \leq |\vec{k}_{2\mathrm{d}}| \leq 1$. $U^\mathrm{nucl}$ is a continuous but not smooth function of $\vec{k}_{2\mathrm{d}}$, while it is likely that $U^\mathrm{nucl}(\vec{k}_{2\mathrm{d}})$ together with the Hamiltonian of Chern insulator $H_\mathrm{Chern}$ can be slightly deformed homotopically so that $U^\mathrm{nucl}$ becomes smooth. In the main text, we assume that $U^\mathrm{nucl}(\vec{k}_{2\mathrm{d}})$ can be taken to be a smooth function of $\vec{k}_{2\mathrm{d}}$.

Similarly, we define the unitary for annihilating a pair of Chern insulators as $U^\mathrm{annih} = X^\mathrm{layer} U^{\mathrm{nucl}\dagger} X^\mathrm{layer}$. As outlined in the main text, the unitary pumping a Chern insulator through a layered system in 3d can be expressed as

$$U_\mathrm{Chern} = \prod_{1 \leq j \leq N} U^\mathrm{annih}_{2j,2j+1} \prod_{1 \leq j \leq N} U^\mathrm{nucl}_{2j-1,2j}, \tag{97}$$

which can also be defined in the presence of $C_2$ twisted boundary condition as discussed in the main text. In either case, each of $U^\mathrm{nucl}, U^\mathrm{annih}$ does not commute with $Z^\mathrm{fl}$ due to the presence of the factor that depends on $Y^\mathrm{fl}$, but they cancel out with each other in the expression of $U_\mathrm{Chern}$. $U_\mathrm{Chern}$ after all preserves the flavor index, and acts within the Fock space of each flavor $\alpha = 0, 1$ with $Z^\mathrm{fl} = (-1)^\alpha$. Each unitary acting within one specific Fock space for a flavor is written as $V_{\alpha=0}, V_{\alpha=1}$, each of which is identified as pumping a p+ip superconductor.

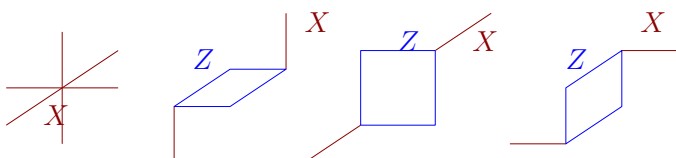

Figure 6: The Hamiltonian terms for fermionic toric code model on cubic lattice.

## D  Fermionic Toric Code in (3+1)D

Let us review the fermionic toric code in (3+1)D [63], whose ground states describe $\mathbb{Z}_2$ gauge theory with emergent fermion.

It is more convenient to describe the theory using dual $\mathbb{Z}_2$ 2-form gauge field instead of 1-form gauge field. In terms of dual $\mathbb{Z}_2$ 2-form gauge field $b$, which satisfies $db = 0$ mod 2, the theory for the ground states has the topological term

$$\pi \int b \cup b \ . \tag{98}$$

To see this is the correct topological term, we can rewrite it as $\pi \int b \cup w_2$ where $w_2$ is the second Stiefel-Whitney class for the tangent bundle. We can restore the 1-form gauge field as a Lagrangian multiplier that enforces $db = 0$ by the action $\pi \int b \cup da$. Then integrating out $b$ instead enforces $da = w_2$, which implies that the Wilson line of $a$ is a fermion.

Let us construct a lattice model for the action. It can be obtained by gauging $\mathbb{Z}_2$ 1-form symmetry in $\mathbb{Z}_2$ 1-form symmetry protected topological phase with the cocycle $\phi(b) = b \cup b$, the latter is discussed in [100]. We introduce qubit on each face, acted on by the Pauli operators $X_f, Y_f, Z_f$. The result is the Hamiltonian

$$H = -\sum_e \prod_{f:e\in\partial f} X_f \cdot (-1)^{\int b\cup\tilde{e}+\tilde{e}\cup b} - \sum_c \prod_{f\in\partial c} Z_f \ , \tag{99}$$

where $e$ are edges, $f$ are faces and $c$ are cubes, $\tilde{e}$ is the 1-cochain that takes value 1 on an edge $e$ and 0 otherwise, and $Z_f = (-1)^{b(f)}$, where $b(f) = 0, 1$ is the value of the 2-form gauge field on face $f$.

The above construction works on any lattice with a triangulation. For instance, we can consider the Hamiltonian model on lattice with the topology of $\mathbb{RP}^3$ by suitable twisted boundary condition.

For the special case of cubic lattice, we sketch the Hamiltonian terms in Figure 6 on the dual lattice, where we dualize the surface variable $\hat{Z}_{\hat{f}} = (-1)^{b(f)}$ into edge variable $X_e$ one the dual lattice. The magnetic flux membrane operator is unmodified compared to the ordinary toric code, while the Wilson line needs modification with additional $X$ in order to commute with the plaquette terms, and the modification gives the particle nontrivial statistics.

## E  Detailed descriptions of the partial rotation

Here we review the properties of the partial $C_M$ rotation taken for a disk subregion of a (2+1)D fermionic invertible phase, following the argument of [94, 95].

The partial $C_M$ rotation of the (2+1)D fermionic invertible phase within a disk can be analytically computed by using the cut-and-glue approach established in [101], which describes the entanglement spectrum of the disk subregion in the long wavelength limit by that of the (1+1)D CFT on its edge. That is, the reduced density matrix for the disk subregion $D$ is effectively given by $\rho_D = \rho_{\mathrm{CFT}}$, where $\rho_{\mathrm{CFT}}$ denotes the CFT on the edge of the disk. The edge of the disk entangled with the complement subsystem is described by a thermal density matrix of a perturbed edge CFT at high temperature [102]. The form of the perturbation in the entanglement Hamiltonian is not universal. In the following, we assume that the entanglement Hamiltonian is that of the unperturbed CFT: $\rho_{\mathrm{CFT}} = e^{-\beta H}$, where the validity of this assumption should be checked with numerics.

Due to the crystalline equivalence principle [91, 92], the $C_M$ rotation symmetry effectively acts as a translation symmetry combined with an internal $\mathbb{Z}_M$ symmetry of the edge CFT on the boundary of $D$. For simplicity, here we assume that the $\mathbb{Z}_M$ internal symmetry acts trivially on the state. See [95] for the discussions about the case with the non-trivial $\mathbb{Z}_M$ action, where the partial rotation defines the invariants of the crystalline invertible phases.

The partial rotation then reduces to evaluating the expectation value of the translation operator within the (1+1)D edge CFT. Let us write the state of a (2+1)D fermionic invertible phase as $|\Psi\rangle$, with chiral central charge

$c_-$. The partial rotation is then expressed as

$$\langle\Psi|\, C_M[D^2]\, |\Psi\rangle = \frac{\mathrm{Tr}[e^{i\tilde{P}\frac{L}{M}}e^{-\frac{\xi}{v}H}]}{\mathrm{Tr}[e^{-\frac{\xi}{v}H}]}$$

$$= e^{-\frac{2\pi i}{24M}c_-}\frac{\sum_{a=1,\psi}\chi_a\left(\frac{i\xi}{L}-\frac{1}{M}\right)}{\sum_{a=1,\psi}\chi_a\left(\frac{i\xi}{L}\right)} \tag{100}$$

where we introduced the velocity $v$ of the CFT, finite temperature correlation length of the edge theory $\xi := \beta v$, the length of the boundary $L = |\partial D|$. $\tilde{P}$ is the normalized translation operator

$$\tilde{P} := \frac{1}{v}(H - E_0) = \frac{2\pi}{L}\left[L_0 - \frac{c_-}{24} - \langle L_0 - \frac{c_-}{24}\rangle\right] \tag{101}$$

so that $\tilde{P}\,|\mathrm{vac}\rangle = 0$ on the vacuum state $|\mathrm{vac}\rangle$ of the CFT.

Also, $\chi_a(\tau)$ is the chiral Virasoro character of $\mathrm{Spin}(2c_-)_1$ WZW model that corresponds to the partition function on a torus. That is,

$$\chi_a(\tau) = \mathrm{Tr}_a[e^{2\pi i\tau(L_0-\frac{c_-}{24})}] \tag{102}$$

where $a$ labels the chiral primary field, which is valued in $\{1,\psi,\sigma\}$ when $c_- \in \mathbb{Z}+1/2$, and valued in $\{1,\psi,v,v'\}$ when $c_- \in \mathbb{Z}$.

Let us evaluate the above CFT characters in the case with even $M$. We perform the modular $ST^MS$ transformation on the character as

$$\sum_{a=1,\psi}\chi_a\left(\frac{i\xi}{L}-\frac{1}{M}\right) = \sum_{a=1,\psi}\sum_b S_{ab}\chi_b\left(-\frac{1}{\frac{i\xi}{L}-\frac{1}{M}}\right)$$

$$= \sum_{a=1,\psi}\sum_b (ST^M)_{ab}\chi_b\left(\frac{-iM\frac{\xi}{L}}{\frac{i\xi}{L}+\frac{1}{M}}\right) \tag{103}$$

$$= \sum_{a=1,\psi}\sum_{b,c}(ST^M)_{ab}S_{bc}\chi_c\left(\frac{iL}{M^2\xi}+\frac{1}{M}\right)$$

By plugging the modular $S,T$ matrices of $\mathrm{Spin}(2c_-)_1$ WZW model into the above expression, we get

$$\sum_{a=1,\psi}\chi_a\left(\frac{i\xi}{L}-\frac{1}{M}\right) = e^{-\frac{2\pi iM}{24}c_-}\sum_c\chi_c\left(\frac{iL}{M^2\xi}+\frac{1}{M}\right) \tag{104}$$

At high temperature we have $\frac{L}{\xi}\gg 1$, where we can approximate the character in terms of the highest weight state $|h_c\rangle$:

$$\chi_c\left(\frac{iL}{M^2\xi}+\frac{1}{M}\right) \approx e^{\frac{2\pi i}{M}(h_c-\frac{c_-}{24})}e^{-\frac{2\pi L}{M^2\xi}(h_c-\frac{c_-}{24})}\ . \tag{105}$$

Due to the exponentially dropping factor $e^{-\frac{2\pi L}{M^2\xi}(h_c-\frac{c_-}{24})}$, the lightest state with $c=1$ has the dominant contribution at large system size. We hence get to the leading order

$$\sum_{a=1,\psi}\chi_a\left(\frac{i\xi}{L}-\frac{1}{M}\right) = e^{-\frac{2\pi iM}{24}c_-}e^{-\frac{2\pi i}{M}\frac{c_-}{24}}e^{\frac{2\pi L}{M^2\xi}\frac{c_-}{24}} \tag{106}$$

Also, we have

$$\sum_{a=1,\psi}\chi_a\left(\frac{i\xi}{L}\right) = \sum_{a=1,\psi}\sum_b S_{ab}\chi_b\left(\frac{iL}{\xi}\right) \tag{107}$$

where the character can again be approximated by the contribution of the highest weight state

$$\chi_b\left(\frac{iL}{\xi}\right) \approx e^{-\frac{2\pi L}{\xi}(h_b-\frac{c_-}{24})}\ . \tag{108}$$

Hence, to the leading order we get

$$\sum_{a=1,\psi} \chi_a \left(\frac{i\xi}{L}\right) \approx e^{\frac{2\pi L}{\xi}\frac{c_-}{24}} \tag{109}$$

Combining the above results, we obtain the final expression of the partial rotation

$$\langle\Psi| C_M[D^2] |\Psi\rangle = e^{-\frac{2\pi i}{24}(M+\frac{2}{M})c_-} e^{\frac{2\pi L}{M^2\xi}\frac{c_-}{24}} e^{-\frac{2\pi L}{\xi}\frac{c_-}{24}} \tag{110}$$

Similarly, the expectation value of the partial rotation followed by the partial $\mathbb{Z}_2^f$ fermion parity symmetry is given by

$$\langle\Psi| C_M(-1)^F[D^2] |\Psi\rangle = \frac{\text{Tr}[e^{i\tilde{P}\frac{L}{M}}(-1)^F e^{-\frac{\xi}{v}H}]}{\text{Tr}[e^{-\frac{\xi}{v}H}]}$$
$$= e^{-\frac{2\pi i}{24M}c_-} \frac{\chi_1(\frac{i\xi}{L}-\frac{1}{M}) - \chi_\psi(\frac{i\xi}{L}-\frac{1}{M})}{\sum_{a=1,\psi}\chi_a(\frac{i\xi}{L})} \tag{111}$$

In that case, we have

$$\chi_1\left(\frac{i\xi}{L}-\frac{1}{M}\right) - \chi_\psi\left(\frac{i\xi}{L}-\frac{1}{M}\right) = \sum_{b,c}(S_{1b}-S_{\psi b})T_b^M S_{bc}\chi_c\left(\frac{iL}{M^2\xi}+\frac{1}{M}\right) \tag{112}$$

By plugging the modular $S,T$ matrices of $\text{Spin}(2c_-)_1$ WZW model into the above expression, we get

$$\chi_1\left(\frac{i\xi}{L}-\frac{1}{M}\right) - \chi_\psi\left(\frac{i\xi}{L}-\frac{1}{M}\right) = \begin{cases} e^{\frac{2\pi iM}{8}c_-}e^{-\frac{2\pi iM}{24}c_-}\sum_c\sqrt{2}S_{\sigma c}\chi_c\left(\frac{iL}{M^2\xi}+\frac{1}{M}\right) & \text{if } c_- \in \mathbb{Z}+\frac{1}{2} \\ e^{\frac{2\pi iM}{8}c_-}e^{-\frac{2\pi iM}{24}c_-}\sum_{b=v,v'}\sum_c S_{bc}\chi_c\left(\frac{iL}{M^2\xi}+\frac{1}{M}\right) & \text{if } c_- \in \mathbb{Z} \end{cases} \tag{113}$$

To the leading order with $c=1$, for any $c_-$ we get

$$\chi_1\left(\frac{i\xi}{L}-\frac{1}{M}\right) - \chi_\psi\left(\frac{i\xi}{L}-\frac{1}{M}\right) = e^{\frac{2\pi iM}{8}c_-}e^{-\frac{2\pi iM}{24}c_-}e^{-\frac{2\pi i}{M}\frac{c_-}{24}}e^{\frac{2\pi L}{M^2\xi}\frac{c_-}{24}} \tag{114}$$

We then obtain

$$\langle\Psi| C_M(-1)^F[D^2] |\Psi\rangle = e^{\frac{2\pi iM}{8}c_-}e^{-\frac{2\pi i}{24}(M+\frac{2}{M})c_-}e^{\frac{2\pi L}{M^2\xi}\frac{c_-}{24}}e^{-\frac{2\pi L}{\xi}\frac{c_-}{24}} \tag{115}$$

Summarizing, the results of the partial rotation are given as follows:

$$\langle\Psi| C_M[D^2] |\Psi\rangle = e^{-\frac{2\pi i}{24}(M+\frac{2}{M})c_-} \times \chi, \quad \langle\Psi| C_M(-1)^F[D^2] |\Psi\rangle = e^{\frac{2\pi iM}{8}c_-}e^{-\frac{2\pi i}{24}(M+\frac{2}{M})c_-} \times \chi, \tag{116}$$

where the amplitude $\chi = e^{\frac{2\pi L}{M^2\xi}\frac{c_-}{24}}e^{-\frac{2\pi L}{\xi}\frac{c_-}{24}}$ is a non-universal positive value. Taking $M=2$, $c_- = 1,1/2$ produce the expressions Eqs. (56), (59) respectively.

At large system size, a number of numerical results suggest that the phase of the partial rotation converges to the universal value predicted by the CFT computation [94, 95, 103]. Meanwhile, the amplitude $\chi$ is in general sensitive to the microscopic detail of the wave function and is not expected to be simulated by the CFT result $e^{\frac{2\pi L}{M^2\xi}\frac{c_-}{24}}e^{-\frac{2\pi L}{\xi}\frac{c_-}{24}}$. However, we expect that the ratio between the partial rotations

$$\frac{\langle\Psi| C_M[D^2] |\Psi\rangle}{\langle\Psi| C_M(-1)^F[D^2] |\Psi\rangle} \tag{117}$$

has the universal amplitude predicted by the CFT computation. In our case, the above ratio is the pure phase, which implies that the non-universal amplitudes $|\langle\Psi| C_M[D^2] |\Psi\rangle|$ and $|\langle\Psi| C_M(-1)^F[D^2] |\Psi\rangle|$ are expected to converge to the exactly same value at large system size. This expectation has been verified numerically with the Haldane model for a Chern insulator on a honeycomb lattice [104], with the rotation angle $M = 2, 6$. [10]

---

[10]We thank Yuxuan Zhang for sharing unpublished results of this calculation.

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
