# Peer review of "Higher-group symmetry of (3+1)D fermionic $\mathbb{Z}_2$ gauge theory: logical CCZ, CS, and T gates from higher symmetry"

_SciPost Physics_

## Round 1 · Referee Report · Anonymous · 2024-3-24

Strengths

1. Clear explanation of why pumping the p+ip gives a Z8 symmetry (and not Z16).

2. The parallel discussions on the lattice and in the continuum provide complementary viewpoints to the generalized symmetries in this model.

3. Intriguing proposal for quantum computation from the T gate on RP3.

Weaknesses

In some discussions it may not be clear if the authors have in mind the lattice model or the continuum field theory.

Report

The authors study the generalized global symmetries of the 3+1d Z2 gauge theory with an emergent fermion, a prototypical example of a topological order.
Despite the simplicity of the theory, the authors found that the symmetry is extremely rich, consisting of a 3-group with a hidden non-invertible structure when one studies the junctions. The authors employ a combination of techniques in field theory and lattice models to explore these symmetries. In particular, they construct an interesting, hidden Z8 (0-form) symmetry from inserting a p+ip before gauging. They provide an illuminating field theory discussion to verify the operator is order 8, and subsequently write down the explicit lattice operator to verify this result. This is an excellent paper with important results of interest to condensed matter physicists, mathematicians, and field theorists. I recommend the publication without reservation.

Requested changes

1. In the third bullet point in section 2.1, why is the surface operator "electric"? I find it more natural to refer to it as "magnetic" since the Wilson is already taken to be electric in the second bullet point. Also, it is somewhat redundant that these bullet points are repeated at the beginning of section 3.

2. Optional: I am curious about the phase of the 8th power of the Z8 generator in (26). It may depend on the spatial lattice in an interesting way.

---

## Round 1 · Referee Report · Anonymous · 2024-3-30

Report

The paper gives a systematic study of the generalized symmetries in (3+1)D Z2 gauge theory with an emergent fermion. The paper discussed four types of symmetry defects, among which two of them arise from condensation of emergent fermionic line on 2d surface (the Kitaev defect) and 3d volume (the p+ip defect). It discussed carefully the order of these defects, uncovering that there are reductions of the order. It also studied the interplay between various symmetries, showing a 3-group and non-invertible structure. The authors also showed that these symmetries give rise to non-clifford gates (CCZ, controlled S, and T) on the lattice, demonstrated the importance of understanding topological defects and emergent higher symmetries for fault-tolerant quantum computation. This paper is clearly written, and the results are interesting, so I support it for publication.

The authors may consider clarify the following points better before publication:

1. The author explained the 3d defect to be gravitational Chern-Simons theory, as well as p+ip superconductor. However, the former depends on spin structure only, while the latter depends on both the spin structure and an additional Z2 symmetry (note that the classification of p+ip SC is $\Omega^{Spin}_3(BZ_2)=Z_8$ which requires an additional Z2). This is responsible for Z16 vs Z8 difference that the authors has highlighted. The authors may consider to give a sharper discussion on this.

2. When the magnetic surface operator intersects the 3d p+ip operator, there is a majorana zero mode trapped at the intersection. This consequently leads to non-invertible operator-valued associator. However, this interesting result is not reproduced on the lattice side unfortunately as far as I can tell. I wish the authors can show this explicitly on the lattice.

---

## Round 3 · Author Response

We thank the referees for careful reading of the manuscripts and helpful comments on the draft.

---

## Round 3 · List of Changes

1. We added footnote 4 clarifying the meaning of electric versus magnetic surface operator as requested in the first referee report.
2. We added footnote 5 for alternative explanation for the relation between Z8 and Z16 using cobordism group as requested in the second referee report.
3. We added reference to [72] for the lattice model of the domain wall with Majorana zero mode as requested in the second referee report.
4. We fixed a typo in Eq.99 for the Hamiltonian of the fermionic toric code.

---

## Editorial Decision

accepted_in_target_journal